# Efficient hybrid numerical modeling of the seismic wavefield in the presence of solid-fluid boundaries

Chao Lyu [1] ✉, Barbara Romanowicz[1,2], Liang Zhao [3] & Yder Masson[4]

Applying full-waveform methods to image small-scale structures of geophysical interest buried within the Earth requires the computation of the seismic wavefield over large distances compared to the target wavelengths. This represents a considerable computational cost when using state-of-the-art numerical integration of the equations of motion in three-dimensional earth models. "Box Tomography" is a hybrid method that breaks up the wavefield computation into three parts, only one of which needs to be iterated for each model update, significantly saving computational time. To deploy this method in remote regions containing a fluid-solid boundary, one needs to construct artificial sources that confine the seismic wavefield within a small region that straddles this boundary. The difficulty arises from the need to combine the solid-fluid coupling with a hybrid numerical simulation in this region. Here, we report a reconciliation of different displacement potential expressions used for solving the acoustic wave equation and propose a unified framework for hybrid simulations. This represents a significant step towards applying 'Box Tomography' in arbitrary regions inside the Earth, achieving a thousand-fold computational cost reduction compared to standard approaches without compromising accuracy. We also present examples of benchmarks of the hybrid simulations in the case of target regions at the ocean floor and the core-mantle boundary.

Resolution in seismic tomography of the earth's mantle and crust has been progressively improving, in particular with the advent of full waveform inversion (FWI) approaches based on accurate wavefield computations in 3D earth models using the Spectral Element Method (SEM[1,2]).

However, achieving higher resolution remains a significant challenge, especially for geometries where source-station distances are much larger than the minimum wavelength to be resolved. This is due to the considerable cost of the three-dimensional (3D) seismic wavefield computations, which depend on the fourth power of the target minimum period. In the global tomography case, it is also hindered by the uneven global distribution of sources and receivers, factors currently beyond our control.

At the regional scale, FWI has been successfully employed to produce high-resolution 3D seismic velocity images of the Earth's crust and upper mantle in various parts of the world, including Australia[3], East Asia[4], Europe[5], North America[6-8], and South America[9], with adequate data coverage, and relative computational efficiency due to the comparatively small lateral and vertical size of the inversion domains. In these inversions, both the seismic sources and the receivers are located within the area of interest. We will refer to this kind of setting as the SIRI case (Source Inside and Receiver Inside).

[1]Department of Earth and Planetary Science, University of California, Berkeley, CA, USA. [2]Institut de Physique du Globe, Paris, France. [3]Key Laboratory of Deep Petroleum Intelligent Exploration and Development, Institute of Geology and Geophysics, Chinese Academy of Sciences, Beijing, China. [4]University of Pau and Pays de l'Adour, Pau, France. ✉e-mail: lyuchao1988@gmail.com

In this type of study, mostly surface waves are modeled, limiting resolution at depth.

Making use of seismic waves that originate or are recorded outside of the box can significantly improve the lateral and vertical resolution of the structure without the need to enlarge the horizontal dimensions of the model as required in the traditional SIRI setting.

Over decades, geophysicists developed hybrid numerical simulations in engineering mechanics, oil and gas exploration, and ground motion[10–12], with considerations for reducing the computational effort. Different authors have proposed hybrid approaches, making it possible to reduce the computational burden in the case where sources and/or stations are located outside of the target region, by coupling a global solver outside with a numerically local solver inside it. In this kind of approach, termed "Box Tomography" by Masson and Romanowicz[13], the wavefield outside of the target region, or "box", is computed only once in a fixed 1D or 3D reference model, and successive iterations of the model updates are performed only inside the box. Most studies have focused on the case where sources are outside and receivers inside the box, or on the case where sources are inside and receivers outside the box. Following their work[14,15], we will refer to these two scenarios as the SORI and the SIRO cases, respectively.

For example, under the SORI setting, Monteiller et al.[16] coupled the 1D global Direct Solution Method (DSM[17]) with the 3D local solver SPECFEM3D_Cartesian[1]. Their work successfully imaged the deep roots of the western Pyrenees through FWI of teleseismic P waves and their coda[18]. In the SIRO case, Wu et al.[19] implemented an SEM-DSM 3D hybrid method for modeling teleseismic waves with complicated source-side structures, using DSM to calculate the wave propagation from the box boundary to the remote receiver. Under the SORI or SIRO setting, this situation is also relevant for detailed inversion of source parameters[20,21]. Note that using a 1D global solver, as is frequently done, results in more affordable low-period global computations than using a 3D global solver. However, neglecting the 3D background (i.e. "global") seismic structure may introduce some significant errors in the resulting images within the box. Following the formalism proposed in a series of papers[13,22,23], Clouzet et al.[24] combined the 3D global solver SPECFEM3D_GLOBE[2] with the 3D local solver RegSEM[25], to image upper-mantle radial anisotropy structure beneath North America, using SEMUCB_WM1[26] as global 3D reference model outside of the region.

Different authors also proposed hybrid methodologies for the imaging of regions in the deep earth for which both sources and receivers are outside the target region (referred to in what follows as the SORO case). Wen and Helmberger[27] utilized a 2D numerical finite-difference method (FDM[28]) in a localized region near the Core-Mantle Boundary (CMB) and an analytically generalized ray theory (GRT) in the 1D Preliminary Reference Earth Model (PREM[29]) outside of it. Lin et al.[30] combined the 2D SEM in a localized domain near the CMB and the 2D global solver SPECFEM2D_GLOBE applied to the AK135[31] 1D reference Earth model, but without extrapolation from the boundary of the box to the receiver. Kawai and Geller[32] introduced an approach where seismograms are time-shifted to account for the effect of 3D structure outside of a target region located at the base of the mantle and applied this to image several such regions[33]. Pienkowska et al.[34] combined the SPECFEM3D_Cartesian with the AxiSEM-generated Instaseis databases[35,36] in a 1D Earth background model. Recently, Adourian et al.[15] further extended the Box Tomography approach[13] to the SORO case, with SEMUCB_WM1[37] as the 3D global reference model, SPECFEM_GLOBE as the 3D global solver, and RegSEM[25] as the 3D local solver. In a recent study, Li et al.[38] modeled the 3D structure of the Ultra-Low Velocity Zone (ULVZ) associated with the Hawaiian mantle plume down to a period of 3 s, using AxiSEM3D[39] both outside and inside the target region. This solver assumes a smooth structure in the direction perpendicular to the vertical plane containing the source and the receiver, with all numerical simulations conducted throughout the entire Earth model. Due to the substantial computational demands at low periods, only four local ULVZ models were tested.

A remaining challenge is to implement the Box Tomography approach in the presence of a solid-fluid interface within the target region. A similar challenge exists for oceanic regions near the earth's surface, as would be the case in studies exploiting waveform data from stations located on islands, on the ocean floor, or above it[40–43]. Even though the local solver SPECFEM3D_Cartesian includes solid-fluid coupling, the corresponding hybrid solution with a 3D background earth model and a target region containing solid-fluid interfaces was not implemented in the study of Pienkowska et al.[34], nor in the study of Adourian et al.[15].

Here, we focus on implementing and validating a method for computing the 2D/3D seismic wavefield in the context of a hybrid numerical simulation with a localized domain containing a solid-fluid coupling interface (subsequently referred to as the hybrid solid-fluid coupling simulation, HSFC). In particular, we propose a unified formalism for the displacement potential in the acoustic wave equation that enables HSFC simulations in scenarios where both the source and the receiver are located outside the target domain (SORO case). We present a series of numerical experiments to validate the accuracy of the proposed method. We also explore the convergence, efficiency, and waveform completeness of the HSFC.

## Results
### Problem setting and theoretical approach
In Fig. 1, we illustrate two canonical scattering problems involving HSFC. The comprehensive workflow for solving the HSFC is outlined in Fig. 2. In the Methods section, we outline various components essential for its implementation. We introduce the elastic and acoustic wave equations, expressing the latter using two distinct definitions of the displacement potential. Following that, we illustrate the logical relationship between these two definitions, allowing for the subsequent hybrid simulations to be expressed within a unified framework. Then, we present the nomenclature and the workflow related to the HSFC in the SORI and SORO cases, after which we present the solid-fluid coupling equations based on the two different displacement potential definitions. We derive the corresponding mathematical expressions of the hybrid input and output mirror forces used in the HSFC. Finally, we provide a brief description of the absorbing boundary conditions adopted in our approach.

To verify the validity of HSFC, we conduct several 2D and 3D numerical experiments, using the 1D PREM[29] and 3D SEMUCB_WM1[37] models as global reference models, successively, and we consider two cases, with a box containing the CMB or the ocean floor, respectively. In the following 2D simulations, both the global and local simulations are performed using a SEM solver, SPECMAT (Spectral Element Method in Matlab, Lyu et al.[14]), with identical spatial mesh and time steps in the global and local numerical simulations. This ensures that there is no error introduced during the spatial and temporal interpolations of the hybrid input mirror forces from the global simulation to the local simulation, resulting in hybrid waveforms with minimal error. However, in the 3D simulation, to better represent realistic scenarios, we use the SPECFEM3D_GLOBE solver for the global simulations and the SPECMAT solver for the local hybrid simulations. Additionally, the global meshing and time steps are different in the 3D cases. As a result, we observe larger waveform errors compared to the 2D simulation partially due to different intrinsic spatial and temporal dispersion errors (refer to the discussion section for how to reduce these errors for the case when global and local meshes do not match).

### 2D HSFC at the Core Mantle Boundary
To verify the proposed algorithm in 2D, we construct three local models inside the local domain: the reference model, a model with an ultra-low velocity zone (ULVZ) above the CMB, and a model with an

(a) Source Outside - Receiver Inside

(b) Source Outside - Receiver Outside

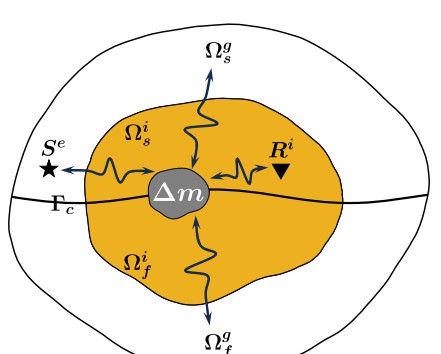
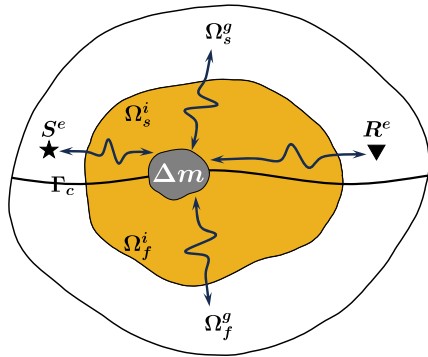

**Fig. 1 | The two canonical setups considered for constructing a general solution to the scattering problem with a solid-fluid interface. a** The source $S^e$ and receiver $R^i$ are located outside and inside the box, respectively. The superscripts $g$, $e$, and $i$ on $\Omega$ represent the global, external, and internal domains. **b** Both the source $S^e$ and the receiver $R^e$ are located outside the box. The global domain $\Omega^g$ comprises a solid part $\Omega_{s^g}$ and a fluid part $\Omega_{f^g}$, separated by a solid-fluid coupling interface $\Gamma_c$.

The local domain $\Omega^i$, a confined box (yellow) within the global domain $\Omega^g$, contains a local solid domain $\Omega_{s^i}$ and a local fluid domain $\Omega_{f^i}$, with a scattering object of interest ($\Delta m$). Leveraging the methodology derived in this paper, after initial global wavefield computations, synthetic seismograms can be computed from an external source to both external and internal receivers by modeling wave propagation only within a compact box that contains the scatterer(s).

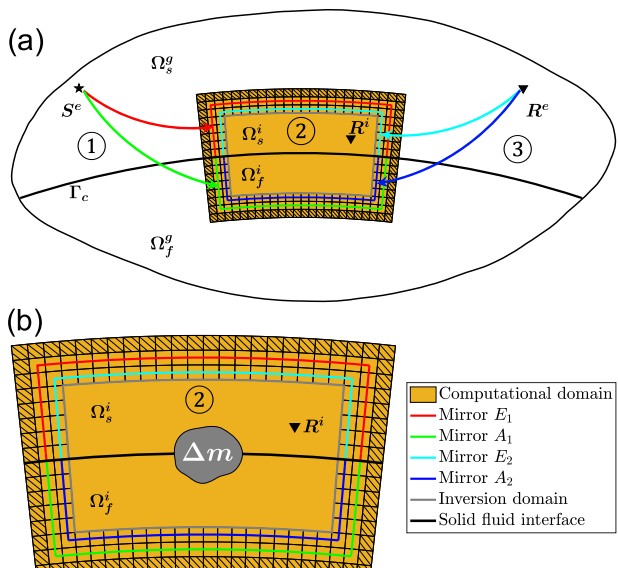

**Fig. 2 | The workflow for calculating hybrid synthetic seismograms from a distant source to a receiver. a** The workflow involves three main steps, as illustrated by the numbered circles. In the initial step, a global simulation is conducted from the remote source to the box, and effective secondary sources are computed and stored at the mirror points $E_1/A_1$. In the second step, the secondary sources are imposed at mirror points $E_1/A_1$ for a local simulation, considering scenarios without a scatterer, and with a scatterer $\Delta m$ (**b**) inside the inversion domain surrounded by gray lines. The resulting hybrid output mirror forces are calculated and recorded at the mirror points $E_2/A_2$. In the third step, two or three global simulations (one per component) are conducted from the remote receiver to the box, generating Green's functions that are subsequently stored at the mirror points $E_2/A_2$. The convolution between the stored hybrid output mirror forces and the Green functions produces the residual synthetic seismograms, capturing the influence of the local scatterer. Elements with black texture are used to absorb the outgoing scattered wavefields.

undulating CMB, as illustrated in Fig. 3a, c, e. The reference "global" model is PREM[29] and includes its lowermost mantle and outer Core structures (Fig. S1a). The vertical (Z) component wavefield of the source-side global simulation is shown in Fig. S2 for the reference model and the Z-component wavefields of the local simulations in the

three local models are shown in Fig. 3b, d, f. In all cases, a single-force point source with 1.667 s dominant period of Ricker wavelet is used. In the case where the local model is the same as the global model, no scattered phases are generated inside the box (Fig. 3b). In the presence of a ULVZ structure located above the CMB, S-waves propagating across the box give rise to a scattered wave resembling a surface wave (Fig. 3d). In contrast, in the case of an undulating CMB structure within the box, the scattered wave exhibits a relatively simpler pattern (Fig. 3f), and the amplitude of the surface wave-like phase is much smaller. Note that the dispersed scattered surface waves are generated by the thin ULVZ anomaly above the CMB. In contrast, the Scholte wave, which has a lower velocity than the S wave, is produced due to the topography of the CMB. Additionally, an S-P scattered phase is generated when the S phase interacts with the topography.

Figure S3 and Fig. 4 show the accuracy of waveforms obtained in hybrid simulations for receivers $R^i$ located inside the box (SORI case) and $R^e$ outside the box (SORO case), respectively. In the SORO case, when the local model is the same as the global reference model, and the local mesh matches the global mesh, the X- and Z-component waveforms obtained for the hybrid simulation at $R^e$ are near zero due to the absence of a model perturbation inside the box (Fig. 4a, b). Figure 4c, d, e, f displays the comparison of waveforms between the global and hybrid simulations in the global and local target models at the receiver $R^e$. The relative errors for the X- and Z-components in the local ULVZ model are approximately 0.013% and 0.004%, respectively. Meanwhile, in the local undulating CMB model, the corresponding errors for the X- and Z-components are approximately 0.573% and 0.232%, respectively. The nearly negligible errors of the SORI (Fig. S3) and SORO (Fig. 4) hybrid simulations demonstrate the effectiveness of our HSFC method. Note that in the 2D SORO cases, the error is smaller than in the 2D SORI cases. The possible reason for this is that the impact of imperfect absorption is smaller in the SORO case than in the SORI case, especially when the global and local numerical simulations have identical spatial mesh and time steps. The corresponding simulations (8 global and 5 local) are shown as animations in Section 7 of the Supplementary Movies 1–13.

## 2D HSFC with ocean, crust, and mantle

Here the reference model is also PREM[29], including its 3 km ocean layer, crust, and mantle. All simulations incorporate the free surface and are performed with the SEM solver SPECMAT. The displacement is recorded at two receivers on the solid side of the ocean-crust boundary, one $R_c^i$ inside and one $R^e$ outside the box, straddling this

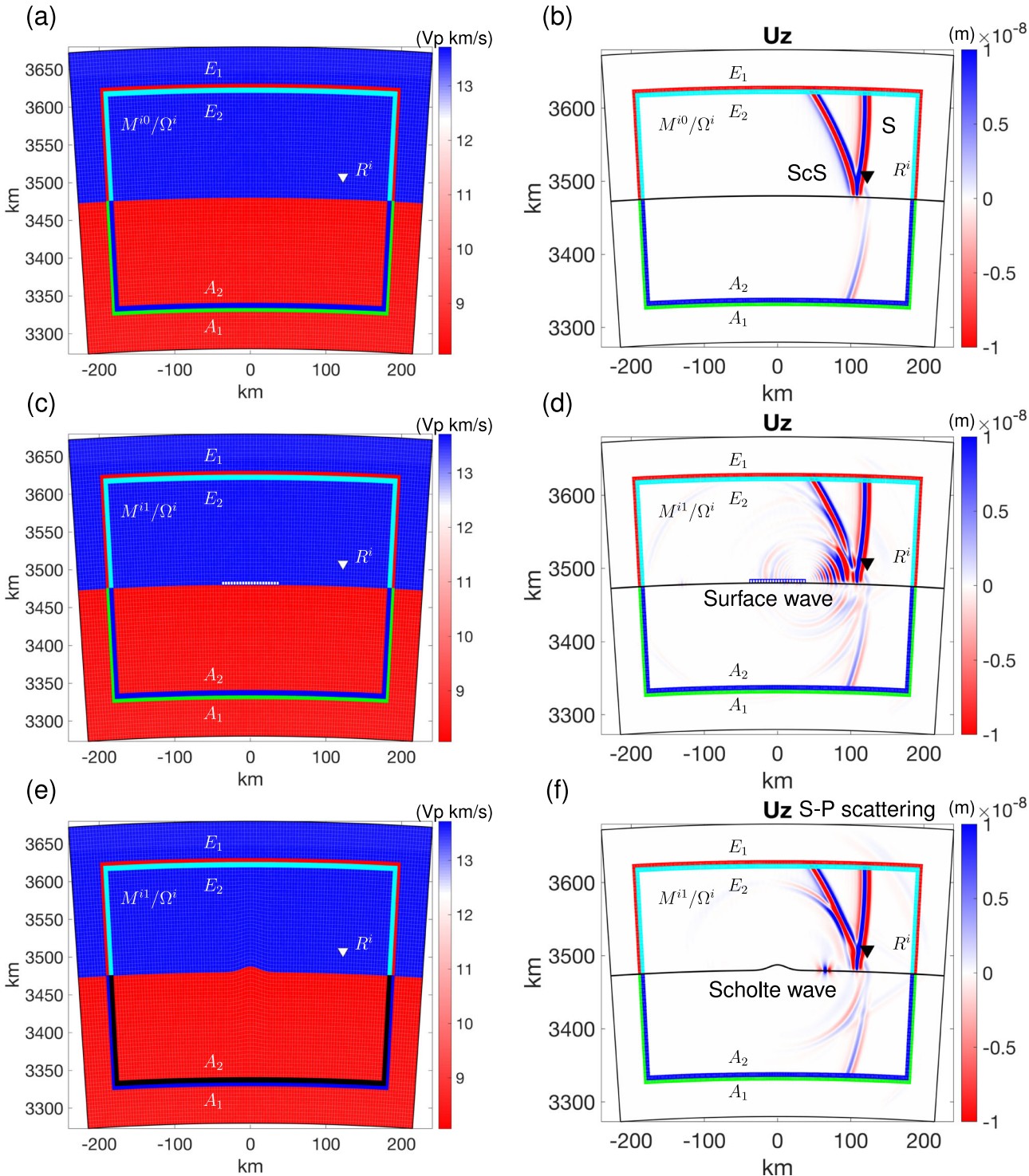

**Fig. 3 | Three 2D local simulations were conducted in three distinct local models with identical hybrid input mirror forces, derived from a global numerical simulation in the global reference model at the corresponding mirror points $E_1$ and $A_1$.** The size of the local domain is (7.5°, 400 km) laterally and it extends 200 km above and below the Core Mantle Boundary (CMB) and the local mesh consists of 120 × (40 + 40) elements. Panels (**a**, **b**) show the local reference model and corresponding wavefield. **c**, **d** show the local target model featuring an Ultra-Low Velocity Zone (ULVZ) and the corresponding wavefield. The ULVZ extends 5 km above the CMB and has a horizontal width of 1.25°, with −30% $V_s$, −10%

$V_p$, and +10% $\rho$ perturbations in elastic parameters, as shown by the mini white block (c) and the blue mesh in (**d**). **e**, **f** same for a local target model an undulating CMB and corresponding wavefield. The Gaussian-shaped undulating CMB is defined by its height (7.5 km) and horizontal width (0.375°). The Gauss-Lobatto-Legendre (GLL) points at mirror $E_2$ within the local elastic domain and at mirror $A_2$ within the local acoustic domain are used to obtain hybrid output mirror forces during the three different local hybrid simulations. 10 elements were used for the absorption at all four boundaries. (b, d, f) are wavefield snapshots at 80 s, also shown as dashed blue lines in Fig. S3.

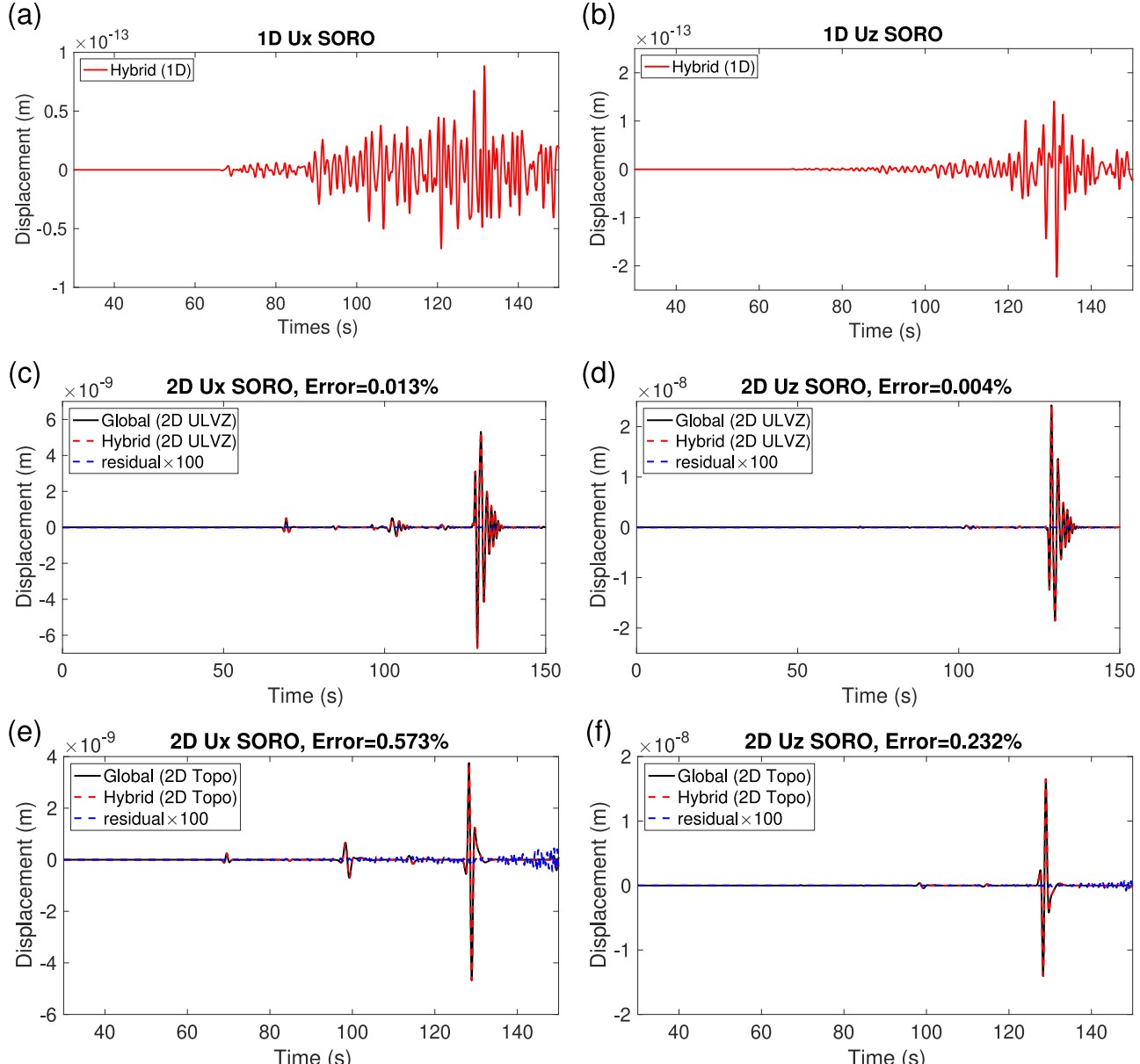

**Fig. 4 | Waveform comparisons in the Hybrid Solid-Fluid Coupling (HSFC) simulation for the Source Outside and Reciever Outside (SORO) cases, at the Core Mantle Boundary (CMB), in 2D. a, b** X and Z component waveforms, respectively, are recorded at outside receiver $R^e$ when the local model is the reference model. These waveforms are expected to have fully zero-values. The solid red lines represent our simulated waveforms following the third-step convolution. The amplitude can be neglected in comparison to the following two SORO cases with local scatterers. **c, d** Same as (**a, b**) for the case where the local model includes an Ultra-Low Velocity Zone (ULVZ) above the CMB. (e, f) same as (c, d) for the case with an undulating CMB within the box. In panels (**c, d, e, f**), the solid black lines correspond to reference global simulations. The dashed red lines correspond to hybrid simulations. Residuals are shown by dashed blue lines, magnified by a factor of 100.

boundary. Eleven additional regularly spaced receivers $R_f^i$ are positioned within the ocean, at a depth of 1 km beneath the free surface, recording the pressure field. We introduce a localized Gaussian-shaped low-velocity structure in the mantle within the local domain. Here, we also maintain the same time step in the local and global simulations, ensuring that no errors are introduced while interpolating the hybrid input mirror forces from the global simulation to the local one.

Figure S4 shows the global reference model and the associated Z wavefield of the global simulation from the remote source $S^e$ side. Figure 5a, c displays the local reference and target models and Fig. 5b, d presents the Z-component wavefields for the local part of the hybrid simulation in both models. S-waves traverse this region at a velocity slower than that of the local reference model in the presence of a

subsurface low-velocity body. Figure S5 displays the Z-component wavefields of two global simulations from the remote receiver $R^e$ side. Figure 6 shows waveform comparisons between global and hybrid simulations for receivers $R_f^i + R_c^i$ (SORI case) and $R^e$ (SORO case). In the SORI case, when the local model matches the global reference model, and their meshes are the same, the hybrid simulation produces nearly identical waveforms to those of the global simulation (Fig. 6e, f and a, b). For the 11 receivers, $R_f^i$ in the ocean, the relative errors of pressure are about 0.047% and 0.302% in the local reference and target models, respectively (Fig. 6e, f). For the receiver, $R_c^i$, the relative errors for X- and Z-components are approximately 0.722% and 0.565%, in the local reference and target models, respectively, with minor deviations attributable to imperfections in the absorbing boundaries (Fig. 6c, d).

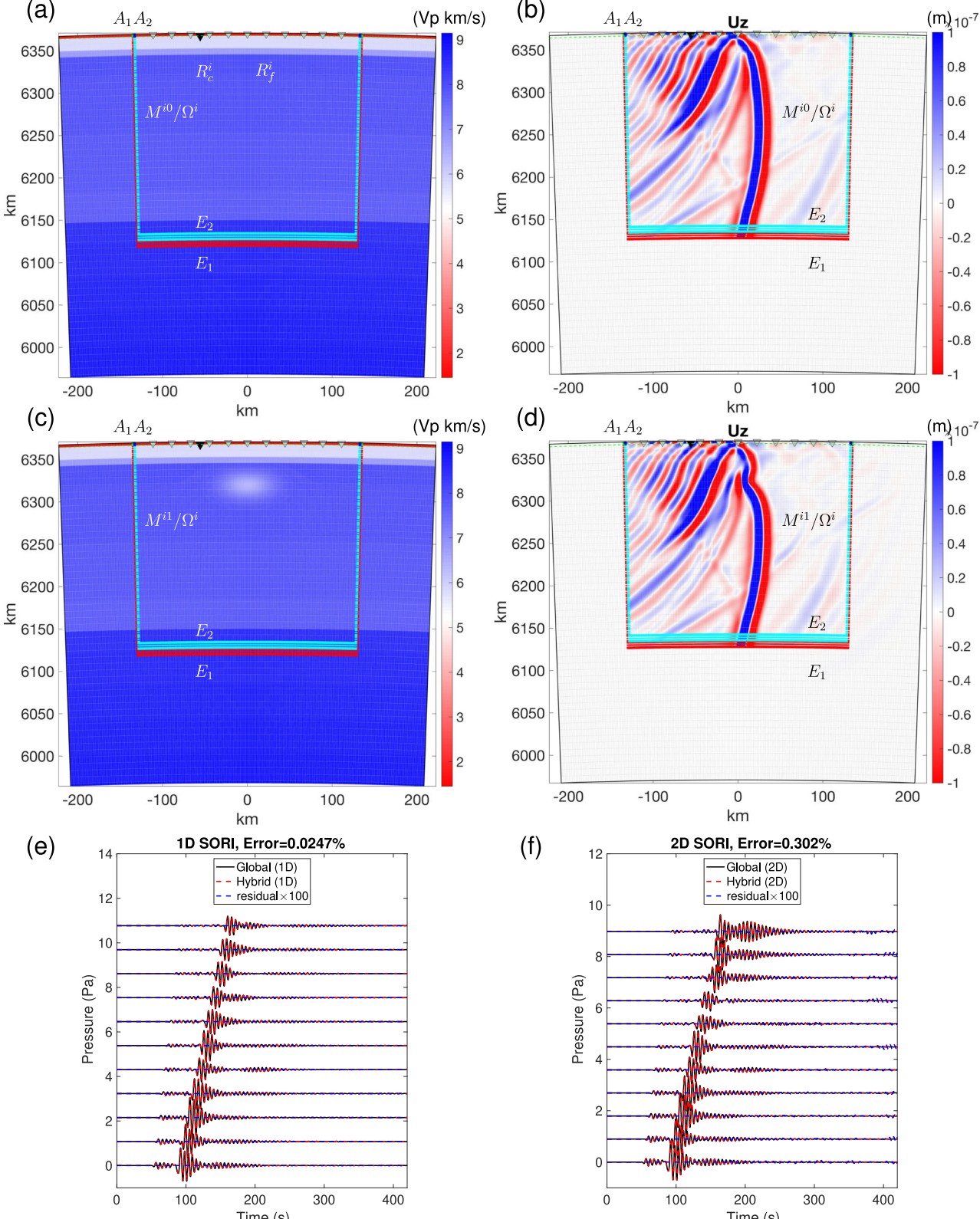

**Fig. 5 | 2D hybrid solid-fluid coupling cases with the ocean positioned above.**
**a**, **b** represent the local reference and perturbed models, respectively. In (**c**), a low-velocity structure (shown by the small white elliptical shape) has been introduced 50 km beneath the free surface, with approximate dimensions of 0.2° × 11.12 km, and − 30%, − 20% and − 10% reductions in Vs, Vp, and density $\rho$, respectively. **b**, **d** Local wavefields obtained in the hybrid simulations, starting from input mirror forces recorded at mirrors $E_1$ and $A_1$ and computed in the global simulation from

the remote source (Fig. S4), which is the same regardless of the local model. The hybrid output mirror forces are calculated at mirrors $E_2$ and $A_2$. The 11 receivers $R_f^i$ are located in the ocean and one receiver $R_c^i$ is located on the ocean-crust boundary. **e**, **f** Corresponding pressure waveform comparison in the Source Outside and Receiver Inside (SORI) case for the two models (**a**) and (**c**). Same color convention as in Fig. 4c. The uppermost red layer signifies the ocean.

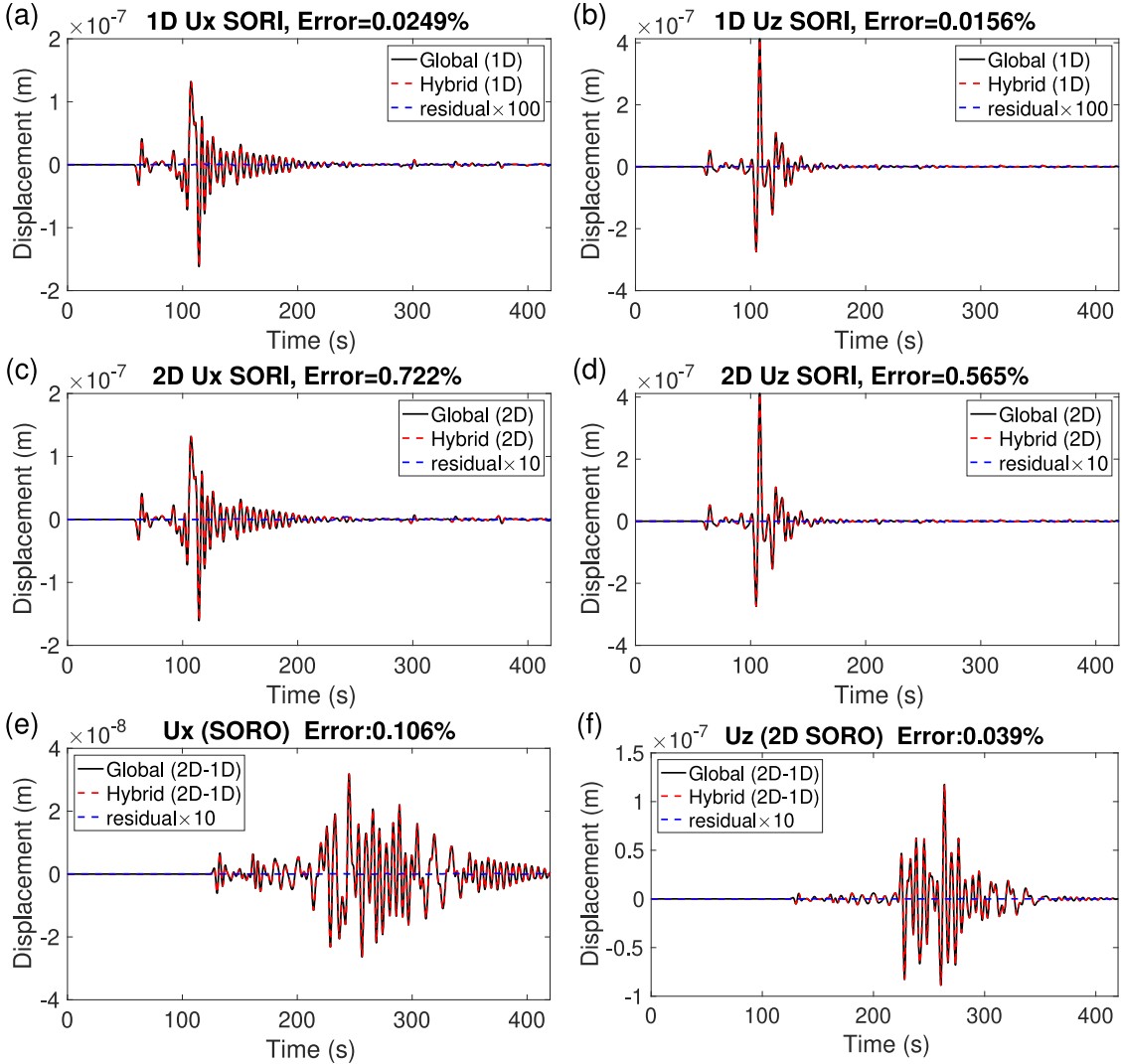

**Fig. 6 | Waveform comparisons for 2D hybrid solid-fluid coupling cases with the ocean.** Source Outside Receiver Inside (SORI, **a**–**d**) and the Source Outside and Receiver Outside (SORO, **e**, **f**) cases for the two models shown in Fig. 5a, c, recorded at the two receivers $R_c^i$ and $R^e$. **a**, **b** The local model is the same as the 1D reference model. **c**–**f** The local model contains a low-velocity structure. Same color convention as in Fig. 4c.

In the SORO case, the relative errors are about 0.1% and 0.04%, for X and Z-components respectively (Fig. 6e, f). Here again, the errors are negligible in both SORI and SORO settings, providing a theoretical foundation for future applications with hybrid simulations that involve the ocean-crust interface.

### 3D HSFC at the Earth's core-mantle boundary

In this 3D example, the background global model is the 3D model SEMUCB_WM1[37] in the mantle, and the 1D global reference earth model PREM[29] in the core. We consider a localized box straddling across the CMB and introduce a ULVZ at the CMB on the mantle side. The local target ULVZ model is shown in Fig. 7a and details on the source and station geometry are given in Supplementary. Section 4. A double-couple source is used for the 3D case. Note that we have smoothed the boundaries of the ULVZ to make it easier for accurate calculation in the global SPECFEM3D_Globe solver than with sharp boundaries. One advantage of the hybrid method is that the flexible meshing of the local domain allows us to better honor the geometry of a ULVZ with sharp boundaries, which would be difficult for global meshing. The minimum resolved period of the Heaviside source time function is 15 s. The CFL condition[44] results in different time steps for the global and local

simulations, because of the presence of the thin low-velocity crust at the top of the global Earth model. Consequently, temporal interpolation is needed to transfer the hybrid input mirror forces from the global simulation to the local one, by taking the same Bspline compression/recovery algorithm as in Adourian et al.[15]. The different time steps in the global and local simulations will generate different temporal dispersion errors, which accounts for larger errors when comparing the global and hybrid waveforms than in the previously discussed 2D case.

Figure S6b and Fig. 7b display the Z-component wavefields generated within the box without and with a ULVZ, respectively. As in the 2D case, the hybrid simulation without any scatterers accurately reproduces the regional wavefield, without artificial energy leaving the box when the structure of the local model matches the background model. However, when the ULVZ is present in the local domain, outgoing scattered wavefields are generated in the hybrid simulation (highlighted by the arrow in Fig. 7b). In the waveforms computed in the SORI case, the L1 difference between the global and hybrid simulations is $\approx 1.4\%$ in the model without ULVZ (Fig. 8a), which may be due to differences in element size and time step between the global and local solvers. Note that the entire local wavefield undergoes a

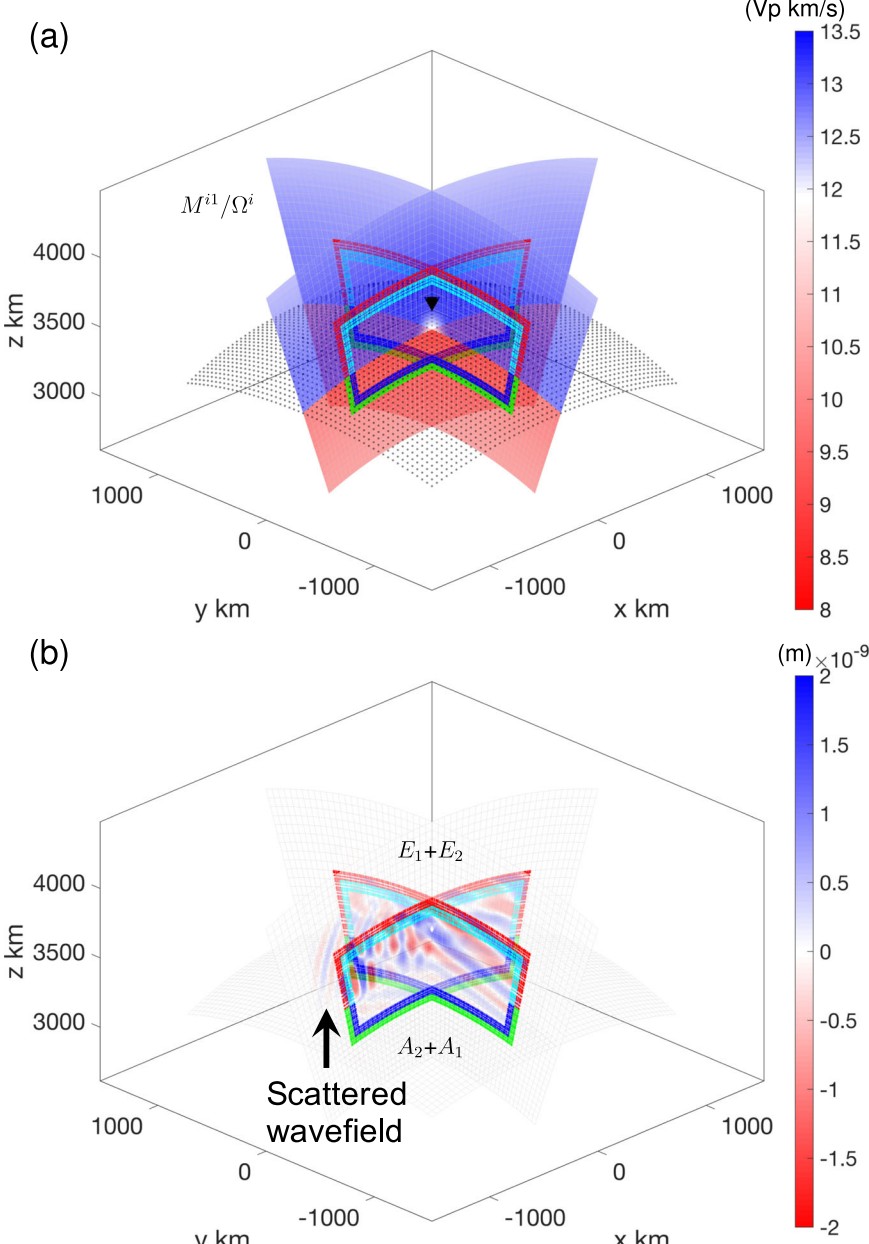

**Fig. 7 | Local 3D target model and hybrid wavefield. a** Local target 3D model includes the lower mantle portion of SEMUCB_WM1 with an Ultra-Low Velocity Zone (ULVZ) (white region at the center of the plot) and the outer core portion of PREM. Black points indicate the Core Mantle Boundary. **b** Corresponding local wavefield on the Z-component at 1260 seconds. Input mirror forces are recorded on mirrors $E_1$ and $A_1$. Output mirror forces and Green's functions are recorded at mirrors $E_2$ and $A_2$ for subsequent convolution. Receiver $R^i$ inside the box is indicated by a black inverted triangle.

transformation into the scattered wavefield after propagating through the mirror $E_1$ and $A_1$ due to the injection of hybrid input mirror forces, and the absorbing layer only works on the scattered wavefield. The incomplete absorption of scattered wavefields in the hybrid simulation with ULVZ results in an increase in waveform errors, approximately around 2% (Fig. 8b). The relatively smaller error in the Z-component is due to its larger waveform amplitude compared to the E and N components (Fig. S7b and Fig. S8b). For the SORO case (Fig. 8c, d), with recording at a distant station, the error is larger than in the SORI case due to the additional convolution operation of the hybrid output mirror forces with Green's functions (see also Supplementary. Section 4). The HSFC produced a post-cursor following the S-phase due to the local ULVZ, and its waveform matches well with the results from the global simulation. Here, the error in the E component (Fig. 8c, d) is smaller than in the N and Z components (Fig. S7c, d and Fig. S8c, d)

because of its larger amplitude. Note that the increase in error at 1150 s in Fig. S7c, d could be due to PcP waves generating an outward-propagating scattered wavefield as they pass through the ULVZ. This scattered wavefield could then be back-propagated to the inside of the Box by the reflection of 660 km and 410 km, as hybrid simulation cannot accurately model secondary scattering. The corresponding Z-component wavefields for the local reference/target models and their residuals are shown as animations in Section 7 of the Supplementary Movies 14–16.

Note that three factors account for the waveform error. i) the global mesh is different from the local mesh, thereby introducing spatial dispersion errors to the hybrid input and output mirror forces. ii) the different time steps used in the global and local simulations introduce different temporal dispersion errors. iii) The third source of errors comes from the imperfect absorbing boundary condition. In

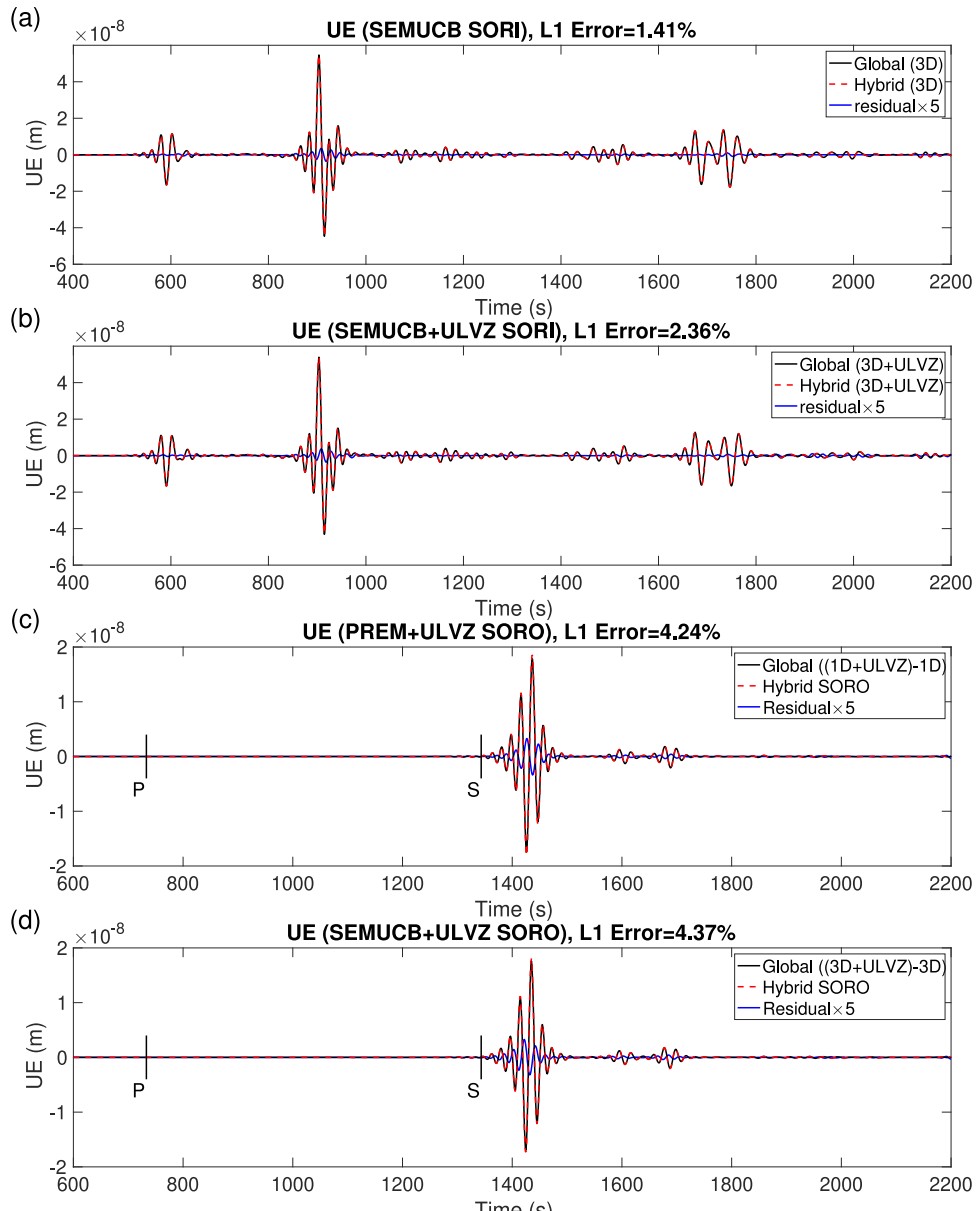

**Fig. 8 | Local waveforms comparison in 3D hybrid simulation with the Source Outside Receiver Inside (SORI) and Source Outside Receiver Outside (SORO) cases. a, b** E-component waveforms at receiver $R^i$ correspond to reference (Fig. S6a) and target (Fig. 7a) models, respectively. **c, d** E-component waveforms at receiver $R^e$ outside the box for the PREM and SEMUCB_WM1 models including the same local Ultra-Low Velocity Zone (ULVZ). Line colors are as in Fig. 4c.

this study, we follow the work in Kosloff and Kosloff[45] and Yao et al.[46]. About 10 absorbing elements assist in absorbing scattered wavefields, enabling us to achieve hybrid solid-fluid coupling. However, especially in 3D problems, they introduce additional computational overhead, and their absorption efficiency is not yet optimal. The Perfect Matched Layer (PML) absorbing condition is very efficient in absorbing the outgoing wavefields, but further development is needed and eventual integration into hybrid numerical simulations, due to its instability in anisotropic elastic models[47]. To minimize spatial interpolation errors, increasing the number of global elements by 1.5 times can be effective, as demonstrated by Lyu et al. (2024). Additionally, utilizing the forward and inverse time-dispersion transforms, as suggested by Lyu et al. (2021) can further reduce time dispersion errors.

## Discussion
We now discuss the following considerations including convergence, waveform integrity, computational efficiency, and the evolving

landscape of global and local solvers of HSFC, in the context of benchmarks performed using the SEM.

## Convergence
To assess the convergence of HSFC, we introduce a smooth ULVZ that can be accurately represented by the various global and local meshes. New 2D global and local models incorporating CMB and a smooth ULVZ are shown in Figs. S9 and S10a1. Discrepancies in spatial dispersion arise from variations between local and global meshing, impacting the computation of hybrid input mirror forces and Green's functions. Spatial mesh for both global and local models is initially defined based on a minimum period of 0.67 s. With this fixed spatial meshing, the minimum period of the Ricker wavelet source is gradually increased. Seven minimum periods of a Ricker source time function, ranging from 0.67 s (1.5 Hz), 0.73 s (1.375 Hz) to 1.33 s (0.75 Hz), are sampled. Snapshots of the wavefield in the local target model at the same moment with different main periods reveal distinct responses to

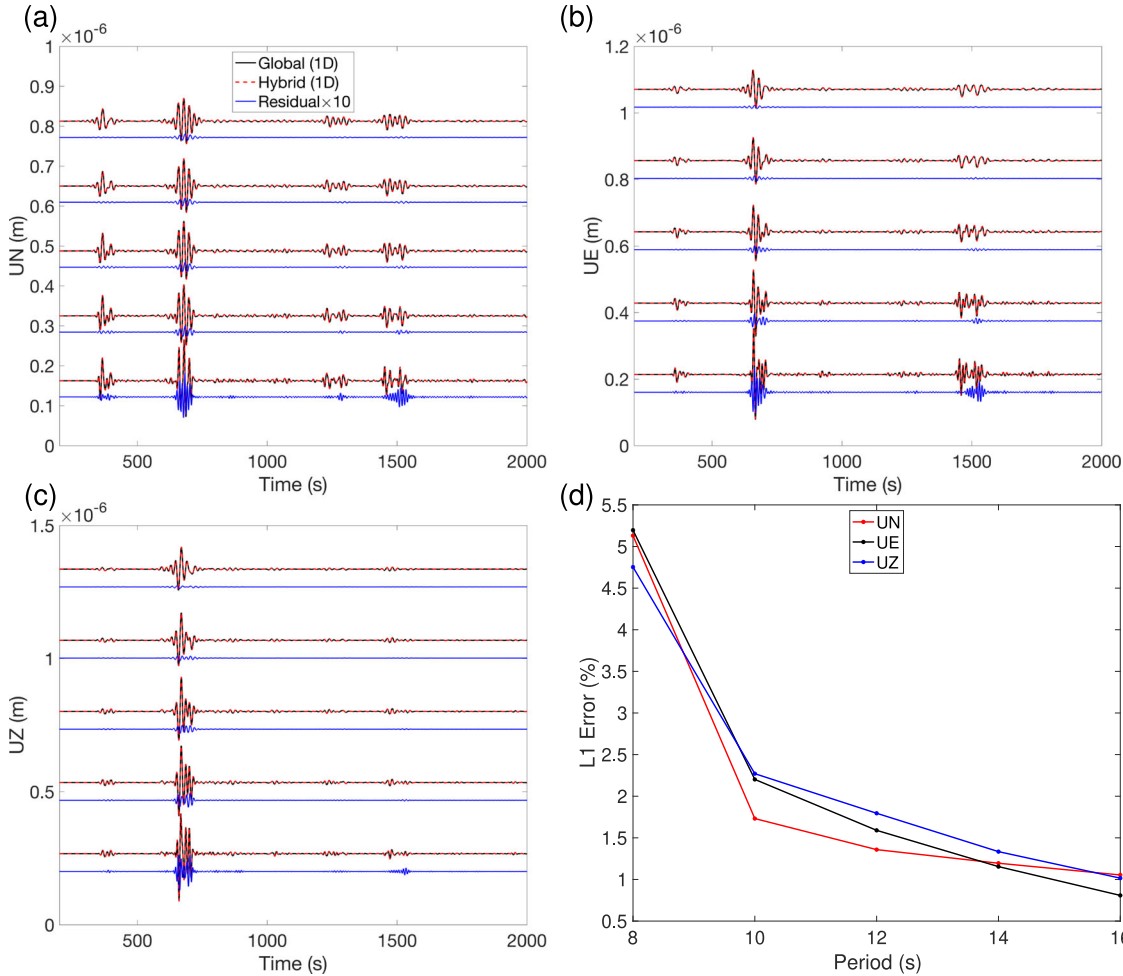

**Fig. 9 | Error in the 3D waveforms as a function of minimum period in the Source Outside Receiver Inside (SORI) case.** Both global and local meshes were determined based on a minimum period of 8 s. Subsequently, seven different sources with minimum periods of 8 s, 10 s, 12 s, 14 s, and 16 s were used to perform the corresponding reference and hybrid simulations. **a**–**c** Comparison of three-component waveforms recorded on the N, E, and Z components respectively, at the same station $R^i$ in Fig. 7a, and the corresponding residuals, magnified by a factor of 10. The corresponding minimum periods increase from the bottom to the top of each plot. **d** Cumulative error of the hybrid simulation, on the 3 components as a function of period.

the same anomaly (Fig. S10, a2 to a8). In the 2D SORI and SORO cases, the waveform errors of the hybrid simulation converge as a function of the period for receivers inside and outside the box (Fig. S10b, c, d, e).

In the 3D case, for the same above local reference model, we fix both global and local meshes based on a minimum period of 8 s. Subsequently, we perform five global and local simulations, respectively, using distinct minimum periods (8 s, 10 s, 12 s, 14 s, and 16 s) of Heaviside source time function. In Fig. 9a, b, c, we display the three-component hybrid waveforms at station $R^i$ inside the box. The corresponding periods are increased from the bottom to the top, revealing the obvious variations in the hybrid waveforms. The associated errors as a function of period are shown in Fig. 9d. The waveform errors converge at about 12 s when we use the global mesh based on the 8s minimum period.

Both 2D and 3D results demonstrate the necessity of increasing the number of elements of the global mesh by approximately 1.5 times the standard for numerical simulation, to achieve very precise hybrid input/output mirror forces, ensuring the accuracy of a hybrid solid-fluid coupling simulation based on SEM, particularly in scenarios where local and global meshing configurations diverge[48]. To address the different temporal interpolation errors in global and hybrid simulations, the forward and inverse time-dispersion transforms can be further utilized[49]. This analysis validates the convergence of our

HSFC method, ensuring the waveform accuracy for future Box Tomography applications everywhere on/inside the Earth using real data using SEM. The corresponding 7 local simulations (main periods from 3.333 s (0.3 Hz) to 1.667 s (0.6 Hz) including the Z-component wavefields of the same target models, are shown as animations in Section 7 of the Supplementary Movies 17-23.

## Waveform integrity

For hybrid numerical simulations, accurately calculating targeted seismic phases is a crucial issue. During the standard global simulation with the long wavelength structures, waves scattered by anomalous bodies create first-order scattering, reaching surface stations. These waves reflect into the global model, causing second-order scattering. However, due to geometric spreading and intrinsic attenuation, second-order scattered waves may not be strong enough to propagate back to the surface stations. In hybrid numerical simulations, first-order scattered waves, generated due to local anomalous bodies, will be absorbed by the absorbing boundary condition, preventing the reflection phases of the scattered waves from returning to the interior of the simulation box, leading to a complete absence of second-order scattering energy for the station outside the box, if the absorbing layers work well. Note that the second-order scattering energy produced by global simulation is weaker than the first-order scattering.

Consequently, in the context of actual deep subsurface structures within a long-wavelength 3D background Earth model, hybrid numerical simulations should be highly applicable. Note that except for the free surface, all the structure discontinuities or large anomalies outside of the box will also generate second-order scattering waves in the standard global simulations. It is better to define a box that contains all the nearby discontinuities and locate it far from known strong reflectors so that the target first-order scattered phase is not affected.

In Section 6 of the Supplementary Info (SI), we've modified the background models with all four boundaries as free boundaries (Fig. S9) and performed a relatively long duration (600 s). Figure S11 shows the corresponding hybrid X- and Z-component waveforms for the SORO case. The second-order scattering phase arrives very late at about 570 s, compared to the first-order scattering at about 130 s, and the amplitude of the second-order scattered waves significantly diminishes due to geometric spreading, becoming much smaller than the amplitude of our target phase (SHdiff's post-cursor). In the real Earth model, we need to further consider intrinsic attenuation, which is not considered here. Therefore, in practical application scenarios, taking into account the large scale of the global long-wavelength Earth model and the effects of geometric and intrinsic attenuation, the amplitude of second-order scattered waves is expected to be significantly reduced in comparison to the primary first-order scattered phases and will exhibit a considerably delayed arrival time compared to the initial first-order scattered phase.

## Efficiency

In the 2D HSFC case, both global and local numerical simulations were conducted on a 2020 MacBook Pro with a 2.4 GHz core and 64 GB memory using MATLAB Version 2023a. A global simulation using SPECMAT takes approximately 33.0 min, while the corresponding hybrid simulation takes about 2.0 min. The nearly 16-fold increase in computational efficiency between global and local simulations, coupled with a consistent ratio of global to local number of elements, suggests a correlation. The additional one minute observed in the global simulation compared to the theoretical factor of 16 is mainly due to the computational demand of calculating a substantial number of hybrid input mirror forces. By adopting different global and local time steps in the 3D case, we achieved even higher computational efficiency compared to the 2D case at the expense of slightly larger waveform errors. The reduction in computational time resulting from the size reduction in the hybrid simulation illustrates its efficiency and underscores the high promise of box tomography.

In the case of 3D HSFC simulations, one global simulation using SPECFEM3D_GLOBE requires approximately 11,857.9 CPU hours on the ANVIL HPC platform[50], compared to 4 CPU hours for the corresponding local simulation within the target region in SPECMAT. The number of elements is 614 times larger in the global than in the local simulation, while the time step is smaller by a factor of 3.56 in the global simulation, due to a thin upper crust (refer to SI. Section 4 for detailed values). The theoretical reduction value, neglecting different computation costs for solid and fluid elements in SEM, is approximately 2185.8 (614 × 3.56). The nearly 3000-fold (11857.9/4) reduction in actual computational time underscores the efficiency and promise of hybrid numerical simulations in Box Tomography. This efficiency increases proportionally with decreasing size of the box. Note that the actual reduction in computational time is larger than the theoretical ratio, due to the nearly 50% fluid domain in volume in the 3D local domain, and it is inversely proportional to the size ratio between the global domain and the local domain. Assuming a consistent reduction ratio with decreasing periods, the significant computational cost reduction of the hybrid simulation will open opportunities for advancing higher resolution in seismic tomography, such as the plug and play (PnP) and image denoisers[51].

## Future developments in global and local solvers

Performing low-period hybrid numerical simulations (minimum period of a few seconds and less), relying on the SPECFEM3D_GLOBE[2] for global simulations in a 3D background model, remains computationally expensive. It may be useful to explore or develop more efficient global numerical solvers. For example, AxiSEM3D[39] and the SALVUS[52] with the anisotropic adaptive mesh refinement offer orders of magnitude faster performance than the SPECFEM3D_GLOBE[2], making them well-suited for calculating hybrid input mirror forces and Green's functions in existing 3D global reference models, at the expense of assuming that the global model is smooth in the direction orthogonal to the vertical plane containing the source and the receiver, as does anisotropic mesh refinement[53]. Recently, Masson[54] proposed a new numerical wave propagation solver, the distributional finite difference method (DFDM), with promising efficiency and flexibility against SEM. In recent work, Masson et al.[55] and Lyu et al.[56] implemented the DFDM approach in spherical geometry, in elastic anisotropic 2D global and 3D regional earth models, respectively, demonstrating its potential for global seismology. Consistent displacement potential definitions can also be formulated to implement hybrid solid-fluid coupling in the target region using the SBP-SAT Finite Difference Method[57]. In addition, the flexible Discontinuous Galerkin Method[58–61], which can naturally handle solid-fluid coupling, are also very promising in the corresponding hybrid seismic-wave numerical simulations. These advancements indicate promising avenues to enhance the efficiency and capabilities of hybrid numerical simulations as presented here. The local solver SPECMAT used here combines several features, including curvilinear mesh, anisotropy, solid-fluid coupling, and absorbing boundary conditions. However, to make the code applicable to real data, implementation of attenuation is needed. Although the code is written in Matlab, the computational efficiency is remarkably high. For a 3D regional model with elements of 56 × 56 × 31 and a total of 8000-time steps, the simulation can be completed in only 4 CPU hours. This high efficiency enables us to run many local simulations simultaneously, allowing for various parameter/structure explorations in the target domain.

## Outlook

This study presents the previously lacking theoretical steps necessary to enable hybrid numerical simulations of the seismic wavefield targeting remote regions that include solid-fluid boundaries, making it possible to image fine-scale structures anywhere within the Earth, given the availability of a sufficiently accurate background global model. Examples of potential applications of Box Tomography in this context are for seismic imaging of complex structures at the base of the mantle such as ultra-low velocity zones[38,62–64], local solid-fluid interface fluctuations at the CMB, or the seafloor. Further parallelization and implementation of the HSFC into regional SEM or DFDM solvers will be necessary for applications at even lower periods (i.e. a few Hz) such as necessary for the study of small-scale structures near the ICB[65–69].

## Methods

In this section, we break down the essential theoretical and methodological steps for implementing the proposed hybrid solid-fluid coupling simulation method.

## Elastic wave equation

The propagation of seismic waves in the solid part of Earth (the Crust, Mantle, and Inner Core) is governed by the equations of motion:

$$
\begin{aligned}
\rho_s \ddot{\mathbf{u}}_s &= \nabla \cdot \boldsymbol{\sigma} + \boldsymbol{f}_s \\
\boldsymbol{\sigma} &= \mathbf{C} : \boldsymbol{\varepsilon} \\
\boldsymbol{\varepsilon} &= \tfrac{1}{2}\left[\nabla \mathbf{u}_s + (\nabla \mathbf{u}_s)^T\right],
\end{aligned}
\tag{1}
$$

where $\mathbf{u}_s(\mathbf{x}, t)$ is the displacement field vector, $\rho_s(\mathbf{x})$ is the density, $\boldsymbol{\sigma}(\mathbf{x})$ is the stress tensor, $\boldsymbol{\varepsilon}(\mathbf{x})$ is the strain tensor, and $\boldsymbol{f}(\mathbf{x}, t)$ are the body forces at point $\mathbf{x}$ in the elastic domain $\Omega_s$; $\mathbf{u}_s$ is subject to boundary conditions (i.e., traction vanishes at the Earth's surface). The double-dot subscript indicates the second derivative in time.

## Acoustic wave equations based on two different displacement potential expressions

In the fluid part of the Earth (e.g. the ocean and the outer core), the propagation of acoustic waves is governed with an irrotational, inviscid, and no gravity effects assumption by:

$$\rho_f \ddot{\mathbf{u}}_f + \nabla P = \boldsymbol{f}_f \tag{2}$$

and

$$P + \kappa \nabla \cdot \mathbf{u}_f = 0 \tag{3}$$

where $\mathbf{u}_f(\mathbf{x}, t)$ is the displacement field vector, $\boldsymbol{f}_f$ is the force vector, $P$ is the pressure, $\rho_f$ is the fluid density, and $\kappa$ is the bulk modulus of the fluid[70]. In general, the lossless acoustic medium is fully described by only two parameters: density $\rho_f(\mathbf{x})$ and speed $V_P(\mathbf{x})$ such that $\kappa(\mathbf{x}) = \rho(\mathbf{x}) V_P^2(\mathbf{x})$. $\mathbf{u}_f$ is subject to boundary conditions (i.e., pressure vanishes at the Earth's Ocean surface).

We assume there are no sources in the fluid domain so that $\boldsymbol{f}_f = 0$ and the displacement can be expressed in terms of a scalar potential, as is done in the popular spectral element codes SPECFEM[2,71]. However, the SPECFEM2D_GLOBE and SPECFEM3D_GLOBE solvers[71] use different displacement potential definitions, most likely because the displacement potential expression in the package SPECFEM3D_GLOBE is easier to handle when gravity is included. To focus on physical effects that are critical for the longer period band (≥100 s), Chaljub and Valette[72] decomposed the displacement in the fluid domain into two scalar displacement potentials. In this study, we focus on relatively lower periods (≤20 s) and have neglected self-gravitation, resulting in a more simplified wave equation. Both approaches result in a fully explicit fluid-solid coupling strategy. Hereafter, we focus on the displacement potential used in SPECFEM2D_GLOBE and develop the corresponding hybrid simulation workflow. Following that, we make use of the relationship between the two different displacement potentials used in SPECFEM2D_GLOBE and SPECFEM3D_GLOBE, so that the proposed workflow of hybrid simulations with solid-fluid coupling can also be used with SPECFEM3D_GLOBE.

The displacement potential $\varphi$ is defined in SPECFEM2D as follows:

$$\mathbf{u}_f \stackrel{\text{def}}{=} \frac{1}{\rho_f} \nabla \varphi. \tag{4}$$

Using this definition in equation (2), it then follows that:

$$P = -\ddot{\varphi}, \tag{5}$$

where $\ddot{\varphi}$ represents the second derivative of $\varphi$ with respect to time. Substituting this expression into equation (3), we obtain the expression of the acoustic wave equation in terms of the first type of displacement potential:

$$\frac{1}{\kappa} \ddot{\varphi} = \nabla \cdot \left( \frac{1}{\rho_f} \nabla \varphi \right). \tag{6}$$

Note that this expression makes it possible to include first-order discontinuities in the acoustic medium. By employing the displacement potential and pressure definitions of equation (4), such first-order discontinuities can be seamlessly introduced while preserving the continuity of potential and pressure.

In contrast, the SPECFEM3D_GLOBE program utilizes a different definition for the displacement potential:

$$\mathbf{u}_f \stackrel{\text{def}}{=} \frac{1}{\rho_f} \nabla(\rho_f \phi), \tag{7}$$

which leads to the following expression for the pressure in terms of displacement potential:

$$P = -\rho_f \ddot{\phi}. \tag{8}$$

By substituting equation (8) into equation (3) and using $\kappa = \rho_f V_p^2$, an alternative displacement potential representation of the acoustic wave equation is obtained:

$$\frac{1}{\kappa}(\rho_f \ddot{\phi}) = \nabla \left( \frac{1}{\rho_f} \nabla(\rho_f \phi) \right). \tag{9}$$

Note that this displacement potential definition ensures continuity in potential but the displacement is discontinuous for velocity structures with first-order discontinuities. However, the assumption is made in the SPECFEM3D_GLOBE code that the outer core is a fluid domain without any internal discontinuities. Therefore, the spectral-element discretization method remains effective in this case. The two displacement potentials in equations (6) and (9) are related by:

$$\varphi = \rho_f \phi. \tag{10}$$

In Table 1, we present the units of physical quantities attributed to the two different displacement potentials, clarifying relationships between various physical quantities. For the verification of the stability of the solid-fluid coupling[73–75], refer to Section 1 of SI.

## Nomenclature and workflow of hybrid solid-fluid coupling simulation

In this subsection, we introduce the nomenclature related to the hybrid solid-fluid coupling simulation from a distant source to receivers situated within or outside of a specified box (SORI/SORO configurations), as also illustrated in Figs. 1 and 2.

- Global domain $\Omega^g = \Omega_s^g + \Omega_f^g$: the overall domain comprising a solid part, indicated by the subscript $s$ and a fluid part, indicated by the subscript $f$, separated by the solid-fluid coupling interface $\Gamma_c$.
- Local domain $\Omega^i = \Omega_s^i + \Omega_f^i$: a subdomain (hereafter, a confined box) within the global domain $\Omega^g$, including the local solid domain $\Omega_s^i$ and fluid domain $\Omega_f^i$.
- External domain $\Omega^e = \Omega_s^e + \Omega_f^e$: the portion of the global domain outside the local domain, including the external local solid domain $\Omega_s^e$ and fluid domain $\Omega_f^e$.
- Absorbing domain $\Omega^a = \Omega_s^a + \Omega_f^a$: the outermost layer in the local domain (shown with specific texture in Fig. 2), necessary to prevent the outgoing scattered waves from returning to the local domain and compromising the accuracy of the hybrid numerical simulation.
- $E/A$ mirrors domain: $E$ mirror surrounds the local elastic region, while $A$ surrounds the local acoustic region. $E_1/A_1$ consists of a layer of spectral elements. $E_2/A_2$ consists of another layer of spectral elements. The mirror forces are loaded or saved at the discrete points (e.g., GLL points in SEM) in the mirror domain.
- Inversion domain $\Omega^\upsilon = \Omega_s^\upsilon + \Omega_f^\upsilon$: a domain inside the localized box where the model can be updated during the box tomography.

Note that $\Omega^i = \Omega^a \cup E_1 \cup A_1 \cup E_2 \cup A_2 \cup \Omega^\upsilon$. In addition to the domain definitions, models are assigned to the corresponding domains as follows. The global reference model $M^{g0} = M_s^{g0} + M_f^{g0}$ is assigned to $\Omega^g$,

**Table 1 | Physical quantities and units of the different displacement potential expressions used in SPECFEM2D_GLOBE and SPECFEM3D_GLOBE**

| Physical quantity | SPECFEM2D_GLOBE + Unit | SPECFEM3D_GLOBE + Unit |
|---|---|---|
| Displacement | $\mathbf{u}_f = \frac{1}{\rho_f}\nabla\varphi$ in $(m)$ | $\mathbf{u}_f = \frac{1}{\rho_f}\nabla(\rho_f\phi)$ in $(m)$ |
| Displacement potential | $\varphi$ in $(kg/m)$ | $\phi$ in $(m^2)$ |
| Pressure | $P = -\ddot{\varphi}$ in $(N/m^2)$ | $P = -\rho_f\ddot{\phi}$ in $(N/m^2)$ |

All of these are presented in italic text.

including the assumed known external model $M^{e0} = M_s^{e0} + M_f^{e0}$ and local reference model $M^{i0} = M_s^{i0} + M_f^{i0}$. The global target model $M^{g1} = M_s^{g1} + M_f^{g1}$ is also assigned to $\Omega^g$ and includes the assumed known external model $M^{e0} = M_s^{e0} + M_f^{e0}$ and the evolving local target model $M^{i1} = M_s^{i1} + M_f^{i1}$. In Box Tomography[13], once the initial forward global simulations from the remote source and receiver sides have been performed in the global reference model $M^{g0}$, forward and backward simulations are exclusively performed in the successively updated local target model $M^{i1}$ in the inversion domain (Fig. 2), while the external model $M^{e0}$ is left unperturbed.

The workflow of the hybrid solid-fluid coupling simulation of the scattering problem from a distant source to a receiver involves three main steps: from the source to the boundary of the box, within the box, and from the box boundary to the receiver, as illustrated by the numbered circles in Fig. 2a. We first compute the wavefield in the global reference model $M^{g0}$ from the source to mirrors $E1/A1$; Once this is done we iterate the computation of the wavefield from mirrors $E1/A1$ to mirrors $E2/A2$ using the local solver in the evolving model $M^{i1}$. Then we calculate the Green's functions from the stations to the mirrors $E2/A2$. Note that the latter takes advantage of the reciprocity theorem[76], which allows us to perform only 3 computations per station. Finally, we convolve the wavefield at $E2/A2$ with the stored Green's functions to reconstruct the total wavefield from source to station.

Figure 2 b shows the case with a local scatter in the box. More specifically, to efficiently simulate the wavefield propagation of the hybrid simulation in a localized domain with a solid-fluid coupling interface, we have extended the previous workflow[15,23] as the following 5 steps.

1. Before performing global simulation, we need to use a local solver to calculate and record the Cartesian coordinates for GLL points on the four mirrors domain $E_1$, $A_1$, $E_2$, and $A_2$ around the local inversion region inside the closed box containing solid and fluid domains, as shown in Fig. 2. Note that the local solver should support the solid-fluid coupling interface.
2. Then we use a global solver to calculate the seismic wavefield generated by the external source $S^e$ (star in Fig. 2) in a global reference model and store the displacement $U$ at each point on mirror $E_1$ and the displacement potential $\varphi$ on $A_1$, to calculate the hybrid input mirror forces (secondary effective sources), as well as the reference waveforms at the receivers $R^i$ and $R^e$ (reverse triangles located inside and outside of the box on the solid side in Fig. 2).
3. Simulate the wavefield inside the box using a regional solver by imposing the secondary effective sources computed in step 2 at mirrors $E_1$ and $A_1$ around the solid and fluid local domains, respectively. Record the displacement $U$ and acceleration potential $\partial_{tt}\varphi$ wavefields on mirrors $E_2$ and $A_2$, calculate the hybrid output mirror forces, and record the complete waveform at the receiver $R^i$ inside the box.

4. Use a global solver to calculate and record the output displacement $U$ (Green's functions) at each point on $E_2$ and the displacement potential $\varphi$ at each point on $A_2$ from the distant receiver $R^e$ in a global reference model using a delta source time function. Two separate Green's functions are computed in 2D (single force in x and z directions, respectively) and three in 3D (single force in x, y, and z directions, respectively).
5. To obtain the residual waveform due to the local scatterer, we finally convolve the hybrid output mirror forces computed in step 3 with the corresponding Green's functions computed in step 4, sum the contributions over all the GLL grid points on mirrors $E_2$ and $A_2$. By adding the convolved time series to the reference seismogram computed in step 2, we could obtain the complete waveform at the external receiver $R^e$ (if the local model inside the box is perturbed).

On the one hand, if there are no anomalous bodies within the local solid/fluid inversion domain, i.e. when model $M^{i0}$ is used as the local model, the hybrid input mirror forces should fully recover the local solid and fluid wavefields, resulting in a zero-value wavefield outside of the local domain. The signal results obtained from convolution in the above step 5 will be a nearly zero-value time series, with some numerical error. On the other hand, if an updated local model $M^{i1}$ is used as the local model, the presence of local anomalies leads to scattered solid and fluid wavefields propagating outside of the inversion domain. In step 3, these scattered wavefields will contribute to generating the hybrid output mirror forces, acting as third sources, to be convolved with Green's functions in step 5. Then we sum the convolved waveform with the reference waveform acquired in step 2 to generate the comprehensive waveform transmitted from a distant source to a remote station. The absorbing layer $\Omega^a$ is significantly important to maintain the accuracy and stability of the hybrid simulation. The implementation of the absorbing boundary layer will be explained in the SI.

Note that the positions of $E_1/A_1$ and $E_2/A_2$ can be either identical or distinct, with the condition that they remain between the local inversion regions and the absorbing layer. In this study, we choose to place $E_2/A_2$ inside $E_1/A_1$, slightly reducing the impact of incomplete absorption boundaries. The two layers ($E_1/A_1$ and $E_2/A_2$) are located outside of the inversion domain, and the absorbing layers are outside of the two layers of $E/A$ mirrors. In the case when the spectral element method is used, as shown in Figure S2 and Fig. 3 both $E_1/A_1$ and $E_2/A_2$ are composed of a single layer of spectral elements with their internal Gauss-Lobatto-Legendre (GLL) points.

Note that a Cartesian reference frame is used in SPECFEM3D_GLOBE, where the x-axis points East, the y-axis points North, and the z-axis points Up. After the convolution in step 5, the three component waveforms will be expressed in the same coordinate system (x, y, z) at each receiver.

## Solid-fluid coupling based on two displacement potential expressions

For a model containing a solid-fluid interface, the normal components of both the displacement and stress should remain continuous across the solid-fluid interface[70], leading to the following coupled $\varphi - \mathbf{u}_s$ system of equations:

$$\rho_s\ddot{\mathbf{u}}_s = \nabla \cdot \boldsymbol{\sigma} + \boldsymbol{f}_s; \quad \boldsymbol{\sigma} = \mathbf{C} : \boldsymbol{\varepsilon}; \quad \boldsymbol{\varepsilon} = \frac{1}{2}\left[\nabla\mathbf{u}_s + (\nabla\mathbf{u}_s)^T\right]$$

$$\frac{1}{\kappa}\ddot{\varphi} = \nabla \cdot \left(\frac{1}{\rho_f}\nabla\varphi\right) \qquad (11)$$

$$\mathbf{u}_s \cdot \boldsymbol{n} = \frac{1}{\rho_f}\nabla\varphi \cdot \boldsymbol{n}; \quad \boldsymbol{\sigma} \cdot \boldsymbol{n} = -P\boldsymbol{n} = \ddot{\varphi}\boldsymbol{n},$$

where **n** represents the vector normal to the solid-fluid coupling interface, pointing from the fluid domain toward the solid domain. Following the finite element method, we multiply both sides of the equations by test functions, integrate over the entire domain, and use integration by parts. We thus obtain the weak form of the elastic and acoustic wave equations:

$$\int_{\Omega_s} \rho_s \boldsymbol{w} \cdot \ddot{\mathbf{u}}_s d\Omega_f + \int_{\Omega_s} \nabla \boldsymbol{w} : \boldsymbol{C} : \nabla u d\Omega_s + \int_{\Gamma_c} \boldsymbol{w} \cdot \boldsymbol{\sigma} \cdot \mathbf{n} d\Gamma = \int_{\Omega_s} \boldsymbol{w} \cdot \boldsymbol{f}_s d\Omega_s$$

$$\int_{\Omega_f} w \frac{1}{\kappa} \ddot{\varphi} d\Omega_f + \int_{\Omega_f} \nabla w \cdot \left(\frac{1}{\rho_f} \nabla \varphi\right) d\Omega_f - \int_{\Gamma_c} w \frac{1}{\rho_f} \nabla \varphi \cdot \mathbf{n} d\Gamma = 0.$$

$$(12)$$

Here, $w$ and $\boldsymbol{w}$ are the test functions in the fluid and solid domains, respectively. Because of the definition of **n**, a positive sign appears in front of the surface integral over the solid area, and a negative sign in front of the surface integral over the fluid region. By enforcing the continuity of the normal component of displacement and stress, naturally embedded within the third term on the left-hand side through surface integrals, we arrive at the weak form of the $\varphi - \mathbf{u}_s$ coupled solid-fluid wave equation system as follows:

$$\int_{\Omega_s} \rho_s \boldsymbol{w} \cdot \ddot{\mathbf{u}}_s d\Omega_s + \int_{\Omega_s} \nabla \boldsymbol{w} : \boldsymbol{C} : \nabla u d\Omega_s + \int_{\Gamma_c} \boldsymbol{w} \cdot \ddot{\varphi} \mathbf{n} d\Gamma = \int_{\Omega_s} \boldsymbol{w} \cdot \boldsymbol{f}_s d\Omega_s$$

$$\int_{\Omega_f} w \frac{1}{\kappa} \ddot{\varphi} d\Omega_f + \int_{\Omega_f} \nabla w \cdot \left(\frac{1}{\rho_f} \nabla \varphi\right) d\Omega_f - \int_{\Gamma_c} w \mathbf{u}_s \cdot \mathbf{n} d\Gamma = 0.$$

$$(13)$$

We note that during the numerical simulation of solid-fluid coupling in SEM, we need to exchange the displacement $\mathbf{u}_s$ at the solid-fluid interface in the solid domain with the acceleration potential $\ddot{\varphi}$ in the fluid domain at the same spatial grid points on the solid-fluid interface.

Alternatively, using the second definition of the displacement potential $\phi$, the coupled solid-fluid wave equation system can be described using the second definition of the displacement potential $\phi$ by substituting $\varphi$ by $\phi$ in equations (13):

$$\rho_s \ddot{\mathbf{u}}_s = \nabla \cdot \boldsymbol{\sigma} + \boldsymbol{f}_s; \quad \boldsymbol{\sigma} = \mathbf{C} : \boldsymbol{\varepsilon}; \quad \boldsymbol{\varepsilon} = \frac{1}{2}\left[\nabla \mathbf{u}_s + (\nabla \mathbf{u}_s)^T\right]$$

$$\frac{1}{\kappa}(\rho_f \ddot{\phi}) = \nabla\left[\frac{1}{\rho_f} \nabla(\rho_f \phi)\right]$$

$$(14)$$

$$\mathbf{u}_s \cdot \mathbf{n} = \frac{1}{\rho_f} \nabla(\rho_f \phi) \cdot \mathbf{n}; \quad \boldsymbol{\sigma} \cdot \mathbf{n} = -P\mathbf{n} = \rho_f \ddot{\phi}\mathbf{n},$$

which, after multiplying by a test function and integrating, leads to:

$$\int_{\Omega_s} \rho_s \boldsymbol{w} \cdot \ddot{\mathbf{u}}_s d\Omega_s + \int_{\Omega_s} \nabla \boldsymbol{w} : \boldsymbol{C} : \nabla u d\Omega_s + \int_{\Gamma_c} \boldsymbol{w} \cdot (\rho_f \ddot{\phi})\mathbf{n} d\Gamma = \int_{\Omega_s} \boldsymbol{w} \cdot \boldsymbol{f}_s d\Omega_s$$

$$\int_{\Omega_f} w \frac{1}{\kappa}(\rho_f \ddot{\phi}) d\Omega_f + \int_{\Omega_f} \nabla w \cdot \frac{1}{\rho_f} \nabla(\rho_f \phi) d\Omega_f - \int_{\Gamma_c} w \mathbf{u}_s \cdot \mathbf{n} d\Gamma = 0.$$

$$(15)$$

Note that there is an important difference compared to the previous equation (13). Here, we explicitly include the density of the fluid domain at the solid-fluid interface.

Given that, in SPECFEM3D_GLOBE, the calculations in the outer core rely solely on the second type of displacement potential $\phi$, we can obtain the displacement potential $\varphi$ by multiplying the output displacement potential $\phi$ by the corresponding density at each mirror point of $A_1/A_2$ (e.g. equation (10)). This leads to a unified algorithm, efficiently managing solid-fluid coupling and facilitating hybrid numerical simulations for 2D and 3D cases.

## Mathematical expression of hybrid input and output mirror forces

In this section, we will give the explicit mathematical expressions of the hybrid input and output mirror forces obtained in steps 2 and 3 of the workflow. In what follows, we will use the first definition of displacement potential $\varphi$. To transform the weak form presented in equation (12) into a matrix representation of an ordinary differential equation, we rely on the conventional spectral element discretization and assembly of the system and obtain the same equation as Equation (32) in Komatitsch and Tromp[1] and Equation (23) in Cao et al.[70]. For the global reference model $M^{g0}$, the ordinary differential equation governing the time evolution of the global system can be expressed in a discrete $\phi - \mathbf{u}_s$ formalism:

$$\mathbf{M}_s \ddot{\mathbf{U}} + \mathbf{C}_s \dot{\mathbf{U}} + \mathbf{K}_s \mathbf{U} + \mathbf{A}[\ddot{\boldsymbol{\Phi}}]_{fs} = \mathbf{F}_s$$
$$\mathbf{M}_f \ddot{\boldsymbol{\Phi}} + \mathbf{C}_f \dot{\boldsymbol{\Phi}} + \mathbf{K}_f \boldsymbol{\Phi} - \mathbf{A}^T[\mathbf{U}]_{sf} = \mathbf{0},$$

$$(16)$$

where **U** represents the displacement vector of the solid domain in the global system and encompasses the displacement vector at all grid points within the global solid mesh. Additionally, **Φ** denotes the displacement potential vector of the fluid domain in the global system. Associated global mass ($\mathbf{M}_s$ and $\mathbf{M}_f$), and stiffness ($\mathbf{K}_s$ and $\mathbf{K}_f$) matrices in the solid and fluid domains are defined following the definition in Komatitsch and Tromp[1] and Cao et al.[70]. The matrices $\mathbf{C}_s$ and $\mathbf{C}_f$, are the absorbing matrices of sponge-layer ABC, and the matrices **A** and $\mathbf{A}^T$ represent the solid-fluid coupling operations. The operator $[\mathbf{U}]_{sf}$ is utilized to ensure the continuity of displacement and the operator $[\ddot{\boldsymbol{\Phi}}]_{fs}$ is employed to ensure the continuity of the normal stress along the solid-fluid coupling interface following the equation (13). For details, please refer to the same Equation (23) in Cao et al.[70]. From a physics perspective of implementing the solid-fluid coupling, this means that the normal displacement components are transmitted from the solid domain to the fluid domain, and in turn, normal stress components (pressure) are transmitted from the fluid domain to the solid domain.

Following the approach introduced by Masson et al.[22], and based on four discrete spatial window functions ($w_s^{hi}$, $w_f^{hi}$, $w_s^{he}$, and $w_f^{he}$), as defined in the SI and the discrete elastic and acoustic wave equations, we have constructed the mathematical expression of the solid and fluid hybrid input and output mirror forces:

$$\mathbf{F}_s^{hi} = -\mathbf{W}_s^{hi}(\mathbf{K}_s \mathbf{U}) + \mathbf{K}_s(\mathbf{W}_s^{hi} \mathbf{U})$$
$$\mathbf{F}_f^{hi} = -\mathbf{W}_f^{hi}(\mathbf{K}_f \boldsymbol{\Phi}) + \mathbf{K}_f(\mathbf{W}_f^{hi} \boldsymbol{\Phi})$$

$$(17)$$

and

$$\mathbf{F}_s^{he} = -\mathbf{W}_s^{he}(\mathbf{K}_s \mathbf{U}) + \mathbf{K}_s(\mathbf{W}_s^{he} \mathbf{U})$$
$$\mathbf{F}_f^{he} = -\mathbf{W}_f^{he}(\mathbf{K}_f \ddot{\boldsymbol{\Phi}}) + \mathbf{K}_f(\mathbf{W}_f^{he} \ddot{\boldsymbol{\Phi}}).$$

$$(18)$$

The detailed derivation of equations (17) and (18), is given in the SI. Note that in the formulation of the hybrid output mirror forces, we utilize the acceleration potential to compute the hybrid output mirror forces $\mathbf{F}_f^{he}$ within the fluid domain. This differs from the mathematical expression of hybrid input mirror forces in equation (17), where we employ the potential. Due to the use of the scalar acoustic equations (6) and (9), rather than the vectorized acoustic wave equation (2), and due to the fact that the saved Green's function is not displacement but displacement potential, the hybrid input and output mirror "forces" in the acoustic domain are not real physical forces, but the expressions have the same mathematical form as in the elastic wave equation. In the SI, we conducted a detailed dimension analysis based on different contributions from the solid and fluid sides.

## Data availability

All the parameters related to the numerical simulation have been listed in the Figures and SI. All the movies are provided in the Supplementary Data.

## Code availability

The open-source SPECFEM3D_GLOBAL package used in this study is available at https://github.com/geodynamics/specfem3d_globe. HSFC codes are available at https://figshare.com/articles/code/Efficient_hybrid_numerical_modeling_of_the_seismic_wavefield_in_the_presence_of_solid-fluid_boundaries/26956204?file=49054867

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

## Acknowledgements

Barbara Romanowicz and Chao Lyu acknowledge support from the National Science Foundation under Grant EAR-1758198 and from UC Berkeley core funds. Liang Zhao acknowledges support from the National Natural Science Foundation of China Grants 42488201. Chao Lyu also acknowledges partial support from the National Natural Sci-ence Foundation of China Youth Fund Grant 42004045. Computations were performed on the ANVIL system of Purdue University, funded by the National Science Foundation (NSF), through award 2005632[50]. The authors would very much like to acknowledge Professor Daniel Peter for

the discussion of the different implementations of the solid-fluid coupling in the SPECFEM2D and SPECFEM3D_GLOBE as detailed in https://github.com/SPECFEM/specfem3d_globe/issues/821.

## Author contributions

C.L. and B.R. designed the project. All authors contributed to the discussion at various stages of the project. C.L. developed and validated the methodology and wrote the first draft of the paper, and subsequently worked with B.R. on its final version. L.Z. and Y.M. commented and contributed text to the manuscript.

## Competing interests

The authors declare no competing interests.
