## [Transparent Peer Review file · Nature Communications]

Efficient hybrid numerical modeling of the seismic wavefield in the presence of solid-fluid boundaries

Corresponding Author: Dr Chao LYU

Version 0:

Reviewer comments:

Reviewer #1

(Remarks to the Author)

(Remarks on code availability)

Reviewer #2

(Remarks to the Author)

This manuscript describes a methodology for so-called hybrid simulations where different numerical methods can be combined with each other to model (i) the propagation from source to target, (ii) the interaction of the wavefield with the target and (iii) the propagation from target to receivers. The authors pay particular attention to the case where the target contains a fluid/solid interface and how to ensure that boundary conditions are projected from one solver to the next. A particular use case relate to full waveform inversion where the overburden is assumed to be known and where the properties and structure of the target are inverted for, thereby enabling substantial computational savings.

The literature on hybrid modelling is huge dating back to the 1980's at least. Over the last 20+ years numerous authors have focused on how to introduce wavefields in full wave modelling methods such as FD and SEM methods while minimizing numerical artefacts. However, much of this work is not cited in the manuscript.

Although the manuscript is sound and solid and will be of interest to the larger modelling community I struggle to see what is new in the manuscript and certainly question why the manuscript should be suitable for a high-impact journal such as Nature Communications. Certainly the formulation of consistent displacement potential definitions is useful, and similar results can be obtained using for instance a SBPSAT FD formulations ensuring that a fluid/solid boundary condition in the target region is satisfied. The discussion of absorbing boundary conditions is certainly not new but rather a documentation of the specific modelling code developed.

In summary, I believe this is a sound manuscript of interest to the broader community but that it would be more suitable for a journal such as Geophysical Journal International or JGR: Machine Learning and Computation.

(Remarks on code availability)

Reviewer #3

(Remarks to the Author)

This is a well-written and impactful paper for studying Earth's deep interior.

KEY RESULTS:

The main result concerns a significant improvement of the efficiency of full-waveform methods to image small-scale structures of geophysical interest in the case that the region of interest contains a fluid-solid boundary. Full-waveform inversion is an iterative technique aiming to recover (that is, image) material parameters assuming that the initial model is sufficiently close to the "true" one, and accounting for the presence of additive noise in the data. The authors extend the notion of box tomography from solid to mixed solid and fluid contents, where the iteration is required only inside a "box" of interest. The sources and receivers generating the data are placed outside and, in fact, at far (in the sense of wave lengths) distances from the box.

Box tomography is closely related to hybrid computational techniques for solving the system of equations describing seismic waves. The associated algorithms can be developed on the basis of Schur complements. Even though the local solver SPECFEM3D Cartesian includes fluid-solid coupling, the corresponding hybrid solution with a 3D background earth model and a target region containing solid-fluid interfaces was not developed before. This is the novelty of this paper. Specifically, the authors propose a unified formalism for the displacement potential in the acoustic wave equation in the fluid.

The authors correctly point out the shortcoming that in such hybrid numerical simulations, first-order scattered waves, generated due to local anomalous bodies, will be absorbed by the absorbing boundary condition, preventing the reflection phases of the scattered waves from returning to the interior of the simulation box, leading to a complete absence of second-order scattering energy for stations outside the box.

The formulation is extensively tested in computational experiments in 2 and 3D, both with the box containing the core-mantle boundary (CMB) and the ocean floor. In the 2D experiments, identical spatial meshes and time steps in the global and local numerical simulations are employed providing optimal accuracy. The "convergence" studies are focused on comparisons between hybrid and global solutions. I would like to suggest that the authors check the presence of Scholte waves, for example, in Figure 3.

METHODOLOGY:

In general, the numerical approach the authors propose, in all its details, is correct. The acousto-elastic system of equations is used to model seismic waves. I presume that the authors consider relatively high-frequency waves and thus ignore self gravitation. The self gravitation affects the fluid-solid boundary in a significant way.

I would like to suggest that the authors extend their discussion on related work.

To my understanding, the earlier work by Johan Robertsson and Chris Chapman on "An efficient method for calculating finite-difference seismograms after model alterations" (2000; <https://library.seg.org/doi/epdf/10.1190/1.1444787>) is directly related to box tomography.

It has been recognized that fluid-solid boundaries cannot be dealt with within a purely spectral element method. In fact, such boundaries fit naturally a discontinuous Galerkin approach (Käser and Dumbser 2008, <https://library.seg.org/doi/10.1190/1.2870081>; Ye et al. 2016, <https://academic.oup.com/gji/article/205/2/1267/691117>; Guo et al. 2020, <https://www.sciencedirect.com/science/article/abs/pii/S002199912030406X?via=ihub>) Nonetheless, there is a "fix" for spectral elements in the neighborhood of a fluid-solid boundary.

An early reference to the use of velocity potential and the Finite Element Method is Olson and Bathe, "Analysis of fluid-structure interactions. A direct symmetric coupled formulation based on the fluid velocity potential" (1985, <https://www.sciencedirect.com/science/article/pii/0045794985902263>). By changing the definition of displacement potential, one might think that the authors might lose symmetry of the discrete formulation. It would be helpful to comment on this and explain which inner product / Hilbert space is used in the fluid-solid configuration (equations (12)-(13)). In equation (13), in the spectral element formulation of Chaljub and Valette (2004), the average of the fluxes on the fluid-solid boundaries are taken; why did the authors choose not to do that?

As the details of the spectral elements are omitted (which is very reasonable), it seems that the entire subsection entitled "Mathematical expression of hybrid input and output mirror forces" can be omitted.

SIGNIFICANCE:

The significance of the approach presented in the paper is in making the high-resolution study of the CMB region computationally feasible. Rather than resorting to traditional gradient methods, I imagine that plug and play (PnP, see, for example, the work of Ulugbek Kamilov) and the use of image denoisers will find application in deep interior imaging with seismic waves.

(Remarks on code availability)

The code is well documented and the results are reproducible.

Version 1:

Reviewer comments:

Reviewer #1

(Remarks to the Author)

The authors have successfully addressed the issues raised, and I recommend this work for publication in Nature Communications.

(Remarks on code availability)

I have checked the link for SPECFEM3D code and the code is available with that link.

Reviewer #3

(Remarks to the Author)

I appreciate the detailed response by the authors and all the updates of the manuscript. The manuscript has greatly improved and I have no further comments. I recommend publication in Nature Communications.

(Remarks on code availability)

Replies to the review of the manuscript: "Efficient hybrid numerical modeling of the seismic wavefield in the presence of solid-fluid boundaries"

Chao Lyu, Barbara Romanowicz, Liang Zhao, Yder Masson

We thank all three reviewers for their comments. We have done our best to account for all the comments made and to answer all the questions. Reviewers' questions and comments are in black, and our answers are in blue. Please refer to the manuscript for all the references.

1 # Reviewer

Efficiently simulating seismic wave propagation in full 3-D structures is crucial for accurate seismic imaging of Earth's interior, especially for applications like full waveform inversion. Despite advancements such as AxiSEM, which can model smooth 3-D structures, simulating high-frequency seismic waves globally remains challenging. To address this, the authors in this study introduce a hybrid method for box tomography, significantly reducing computational costs while maintaining accuracy. By conducting global simulations once and limiting multiple expensive simulations to smaller local domains, the authors demonstrate that the computational cost could be reduced by three orders of magnitude. Specifically, the authors focus on the examples of solid-fluid coupling simulations in this study and have conducted well-designed benchmarking against full 3-D simulations. This novel method has tremendous potential for widespread applications, including detailed imaging of Earth's interior features like the oceanic lithosphere, mantle transition zone, lower mantle, and inner core. Any progress on these topics will produce an impact beyond seismology. Thus, I highly recommend publishing this study on Nature Communications with some minor revisions as described below.

Contributions from different error sources. The authors discussed errors from spatial and temporal interpolations and Absorbing Boundary Conditions (ABC). However, a deeper analysis of the contributions from each error source would provide valuable insights. Analyzing and comparing the errors due to different model setups can tell us the contribution of each error source. For example, the results in Fig. 4 use the same mesh, so the errors must come from imperfect ABC;

- In Fig. 8c,d of the SORO case, which error between these sources is larger? If the primary error is from an imperfect ABC, better ABCs would help reduce the error. Conversely, if interpolation errors dominate, is there a solution to reduce the error? Further discussions would enhance the manuscript's comprehensiveness and guide future research endeavors.
- Thanks for your careful review. The error in Fig. 8a (SEMUCB+SORI) is mainly from spatial and temporal interpolation errors (about 1.41%) due to the different global and local meshing and time steps when we reasonably assume the imperfect ABCs only introduce neglected reflection errors here. The error in Fig. 8b (SEMUCB+ULVZ+SORI) increases to 2.36%, the additional 0.95% is mainly from the imperfect ABCs of the scattered wavefield due to the ULVZ. Then the error of Fig. 8c,d further increases to about 4.3%. It is larger than in Fig. 8b (SORI case) due to the much

longer distance for the remote receiver, which is equivalently implemented by the additional convolution operation of the hybrid output mirror forces with Green's functions. So the main error here is still from the interpolation error due to the different global and local meshing and time steps.

There is a distinction between the treatment of time and space in our approach. For time, we employ a least squares fit using a B-spline basis, which allows for efficient compression. In contrast, for space, we rely on interpolation methods. Both types of errors can be theoretically estimated since their convergence rates are known. To reduce the spatial interpolation error of the hybrid simulation, one can increase the number of elements in the global simulation by 1.5 times (Lyu et al., 2024 BSSA). To address the different temporal interpolation errors in global and hybrid simulations, the forward and inverse time-dispersion transforms can be further utilized (Wang et al., 2015 Geophysics). We have cited this paper in the main body of the manuscript. Page 11 and Line 283.

Alternatively, one could adopt a variational approach to obtain the wavefield coefficients (similar to what is done in the Distributional Finite Difference Method) to further minimize the error, rather than simply evaluating values at the GLL grid points. While this would likely lead to improvements, it is not directly related to the method we propose, which focuses on how the mirror sources are transformed.

- Meshing ULVZ. In the 3-D ULVZ simulations, does the ULVZ in Fig. 7 have smoothed structure anomalies? The authors mentioned a smooth ULVZ in Fig. S9 and S10, and I suppose it is the same in Fig. 7. It should be clarified. In addition, it is worth noting that one advantage of the hybrid method is that the flexible meshing of the local domain allows us to better honor the geometry of ULVZ with sharp boundaries, which would be difficult for global meshing.

In the 3-D ULVZ simulations, we do use the smoothed version ULVZ here because it is much easier for us to obtain the reference solution calculated by the global solver SPEC-FEM3D_Globe. We are now mentioning it in the text, on Page 8 and Line 201, "Note that we have smoothed the boundaries of the ULVZ to make it easier for accurate calculation in the global SPEC-FEM3D_Globe solver than with sharp boundaries. One advantage of the hybrid method is that the flexible meshing of the local domain allows us to better honor the geometry of a ULVZ with sharp boundaries, which would be difficult for global meshing."

- The waveform details in Fig. 8 suffer from visibility issues due to the plotting scale, hindering the comprehensive understanding of the results. To address this limitation, I recommend replotting Fig. 8 to better visualize the detailed waveforms.

Thank you for your suggestion. We have redrawn Fig. 8. We removed the zero values before the P-phase and only kept the arrival times of the direct P and direct S phases. Additionally, Fig. 8cd shows the residual waveforms, to illustrate the changes in the remote receiver waveforms caused by the presence of the ULVZ. Refer to the updated Fig. 8.

- $\int_{\Omega_s} \omega \rho_s \ddot{u}_s$ in eq. 12 should be $\int_{\Omega_s} \rho_s \omega \cdot \ddot{u}_s$. The product symbol "·" in the last term of eq. 12 and many other equations (eqs. 13, 14, 15 and 19).

Thank you for your careful review. We have checked all the formulations and ensured that all the corresponding dots have been added.

- For eq. 16, add clarification that the corresponding absorbing boundary conditions are Stacey ABC. Thank You. Page 24 and Line 572, we have added clarification: The matrices C_s and C_f , are the absorbing matrices generated due to the sponge-layer ABC.
- Clarify what types of sources (explosion, double-couple, or what else?) are used in the 2-D (i.e., Fig. 3) and 3-D simulations (i.e., Fig. 7).
Page 6 and Line 141, a single-force source is used for the 2D case; Page 8 and Line 201. A double-couple source is used for the 3D case.

- The colors and plotting in Fig. 4 warrant clarification to improve understanding. I think Fig. 4a,b plots scattering waves, which should be theoretically zeros, as the authors described. But how about Figs. 4c-f? Are they solely scattering waves, or do they combine scattered waves with background 1-D waveforms? If they are scattering waves, it is better to use consistent color in (a)-(f) for hybrid seismograms. Currently, a hybrid seismogram is black in a,b while red in c-f. Furthermore, plotting both background 1-D seismograms and scattering waves for comparison would enhance understanding of ULVZ and CMB topography effects. This comparative visualization would elucidate how these heterogeneities influence wave propagation, providing deeper insights into the study’s findings.

We have changed the color of Fig. 4a,b to red so that the hybrid waveforms are in black and the global waveforms are in red for better comparison.

- Describe the corresponding time of the snapshots in Fig. 3b,d,f and Fig. 5b,d. That would help us better link them with the corresponding seismograms.

We have added the following to the caption of Fig. 3, page 39: "(b, d, f) are wavefield snapshots at 80 s, also shown as dashed blue lines in Figure S3".

- Line 196: replace “minimum period” with “minimum resolved period”. And revise it at other relevant places as well. There are many interesting waveform propagation phenomena in Fig. 3 that deserve further analysis. For example, it would be helpful to label out the arrivals S and ScS in Fig. 3b; Explaining the scattered waves, as marked in the figure shown below, would offer valuable insights into how these waves are generated by the structural heterogeneities.

Thank you for your suggestion, please see the updated Fig. 3 with labels. On Page 6 and Line 148,

we have added: "Note that the dispersed scattered surface waves are generated by the thin ULVZ anomaly above the CMB. In contrast, the Scholte wave, which has a lower velocity than the S wave, is produced due to the topography of the CMB. Additionally, an S-P scattered phase is generated when the S phase interacts with the topography."

- In Section 1 of the support material, the authors justify the use of acceleration potential rather than potential in equation S8 through dimension analysis. This justification is fine, but this can be better explained by revisiting the governing equations. Given the ‘force’ formulas of S5 for the scenario of source-to-SEM box coupling, one would write an analogous formula for the coupling from the SEM box to a remote receiver. Indeed, If SEM is utilized to back-propagate the imposed wavefield on the A2 boundaries to a remote receiver, employing the potential is appropriate. However, if Green’s functions are chosen for convolution, we should use a real ‘force’ term, whereas F_f^{hi} in equation S5 is actually not a ‘force’. Although F_f^{hi} is termed a ‘force’ due to its analogy to its solid media counterpart F_s^{hi} , this analogy exists in their mathematical representations. Physically, they are different (they have different units). Similarly, the two equations of S3 have analogous mathematical representation, but not in physics. Equations 1 and 2, along with their counterparts f_s and f_f , exhibit a more direct analogy in both mathematics and physics. Hence, if convolution is employed to derive the synthetic seismogram at a remote station, using equivalent body ‘forces’ akin to f_f would be natural, which differs from F_f^{hi} . Consequently, F_f^{hi} does not have a unit of ‘force’, and referring to F_f^{hi} as a ‘force’ is somewhat inaccurate in this context. By providing this clarification, the physical and mathematical nuances between F_f^{hi} and equivalent body forces become apparent.
- Thank you for your constructive and thoughtful suggestions. As you mentioned, F_s^{hi} in Eq. S5 is not a ‘force’. Let’s first analyze the elastic part in Eq. 16 of the main text. We can derive that the elastic term KU and the source term F_s have the same dimensions, which are force. In seismology, we generally refer to KU as the internal force, which, combined with the external force F_s from the source, can be used to calculate the acceleration term by dividing by the mass (the inverse of the mass matrix). Therefore, the elastic part of the hybrid input and output mirror forces in Eq. S5 is indeed a physical force term. This is originally because we based our following elastic calculations on Eq. 1.

However, in the acoustic domain, if we base our analysis on the acoustic wave equation (Eq. 2), and during the propagation from the box boundaries (A2) to a remote receiver, Green’s function is the displacement, then all the related hybrid input and output mirror force terms inside the acoustic domain will be the same as in the solid domain. However, in practice, we do not use the vectorized acoustic wave equation (Eq. 2) not only due to its higher computational cost compared to the scalar acoustic wave equation (Eqs. 6 or 9) and but also due to the instability when using SEM to solve Eq. 2.

In this study, we use the scalar acoustic wave equations (Eqs. 6 and 9). The saved Green’s function (during the hybrid simulation and from the box boundary to a remote receiver) is not displacement but displacement potential, so the corresponding hybrid output mirror forces cannot be considered real forces. We agree that referring to F_f^{hi} as a ‘force’ is not physically accurate. Therefore, we have added the following sentence to the main body of the text. Line 588: “Due to the use of the scalar acoustic equation rather than the vectorized acoustic wave equation, and due to the fact that the saved Green’s function is not displacement but displacement potential, the hybrid input and output mirror “forces” in the acoustic domain are not real physical forces, but the expressions have the same mathematical form as in the elastic wave equation.”

- In Section 4 of the support material, it’s mentioned that the density of the Ultra-Low Velocity Zone (ULVZ) is set as increasing by 14.3%. However, it’s essential to clarify that this value is utilized solely for synthetic testing purposes, and we know that observational data supports a higher density for ULVZ.
- While the density of real ULVZs typically shows a positive perturbation, the value is not well constrained, and we have chosen to follow the fixed scaling used in FWI: $\frac{\ln V_p}{\ln V_s} = \frac{1}{2}$ and $\frac{\ln V_s}{\ln \rho} = \frac{3}{2}$, as noted at the end of Section 4 of the Supplementary Information.

2 # 2 Reviewer

This manuscript describes a methodology for so-called hybrid simulations where different numerical methods can be combined with each other to model (i) the propagation from source to target, (ii) the interaction of the wavefield with the target, and (iii) the propagation from target to receivers. The authors pay particular attention to the case where the target contains a fluid/solid interface and how to ensure that boundary conditions are projected from one solver to the next. A particular use case relates to full waveform inversion where the overburden is assumed to be known and where the properties and structure of the target are inverted for, thereby enabling substantial computational savings.

The literature on hybrid modelling is huge dating back to the 1980's at least. Over the last 20+ years, numerous authors have focused on how to introduce wavefields in full wave modeling methods such as FD and SEM methods while minimizing numerical artifacts. However, much of this work is not cited in the manuscript.

- Thank you for your suggestion. We have incorporated some related work into the beginning of the Introduction section, starting at Line 47. Over decades, geophysicists developed hybrid numerical simulations in engineering mechanics, oil and gas exploration, and ground motion, with considerations for reducing computational effort. Here we have cited one paper for each field.
 - Bielak, J. and Christiano, P. (1984), On the effective seismic input for non-linear soil-structure interaction systems. *Earthquake Engng. Struct. Dyn.*, 12: 107-119. <https://doi.org/10.1002/eqe.4290120108>
 - Johan O. A. Robertsson and Chris H. Chapman. (2000), An efficient method for calculating finite-difference seismograms after model alterations. *GEOPHYSICS* 2000 65:3, 907-918. <https://doi.org/10.1190/1.1444787>
 - Chiaki Yoshimura, Jacobo Bielak, Yoshiaki Hisada, Antonio Fernández. (2003), Domain Reduction Method for Three-Dimensional Earthquake Modeling in Localized Regions, Part II: Verification and Applications. *Bulletin of the Seismological Society of America*; 93 (2): 825–841. <https://doi.org/10.1785/0120010252>

Although the manuscript is sound and solid and will be of interest to the larger modeling community I struggle to see what is new in the manuscript and certainly question why the manuscript should be suitable for a high-impact journal such as *Nature Communications*. Certainly, the formulation of consistent displacement potential definitions is useful, and similar results can be obtained using for instance an SBP-SAT FD formulation ensuring that a fluid/solid boundary condition in the target region is satisfied. The discussion of absorbing boundary conditions is certainly not new but rather a documentation of the specific modeling code developed.

- Thank you for your kind suggestion. (1) we have moved the absorbing boundary conditions part into the supplemental material Section. (2) In the Future Developments section, on line 361, we have added the possibility of combining SBP-SAT with the proposed hybrid solid-fluid coupling. "Consistent displacement potential definitions can also be formulated to implement hybrid solid-fluid coupling in the target region using the SBP-SAT Finite Difference Method (Dovgilevich and Sofronov, 2015. <https://www.sciencedirect.com/science/article/pii/S0168927414001068>.)"

In summary, I believe this is a sound manuscript of interest to the broader community but that it would be more suitable for a journal such as *Geophysical Journal International* or *JGR: Machine Learning and Computation*.

- Efficiently simulating seismic wave propagation in 3-D structures is crucial for accurate seismic imaging, especially for full waveform inversion. Despite advances like *Specfem3d.Global*, *Salvus*, and *AxiSEM*, global high-frequency simulations remain challenging. Although hybrid numerical simulation techniques have developed over many years, applying hybrid simulations to solid-fluid interfaces such as the CMB or ICB in global seismology has remained challenging due to the lack of hybrid solid-fluid coupling technology. To address this, we have successfully implemented, for the first time in global seismology, a hybrid method for box tomography containing a solid-fluid coupling boundary. This method significantly reduces computational costs while maintaining accuracy,

utilizing a 3D global background model. By conducting global simulations once and focusing on smaller local domains, we achieve a three-order magnitude reduction in cost. We benchmark solid-fluid coupling simulations against full 3-D simulations, demonstrating this method's potential for detailed imaging of Earth's interior using a 3D background model. We believe this progress could impact seismology and beyond.

3 # 3 Reviewer

This is a well-written and impactful paper for studying Earth's deep interior.

KEY RESULTS:

- The main result concerns a significant improvement of the efficiency of full-waveform methods to image small-scale structures of geophysical interest in the case that the region of interest contains a fluid-solid boundary. Full-waveform inversion is an iterative technique aiming to recover (that is, image) material parameters assuming that the initial model is sufficiently close to the "true" one, and accounting for the presence of additive noise in the data. The authors extend the notion of box tomography from solid to mixed solid and fluid contents, where the iteration is required only inside a "box" of interest. The sources and receivers generating the data are placed outside and, in fact, at far (in the sense of wavelengths) distances from the box.

Box tomography is closely related to hybrid computational techniques for solving the system of equations describing seismic waves. The associated algorithms can be developed on the basis of Schur complements. Even though the local solver SPECFEM3D Cartesian includes fluid-solid coupling, the corresponding hybrid solution with a 3D background earth model and a target region containing solid-fluid interfaces was not developed before. This is the novelty of this paper. Specifically, the authors propose a unified formalism for the displacement potential in the acoustic wave equation in the fluid.

The authors correctly point out the shortcoming that in such hybrid numerical simulations, first-order scattered waves, generated due to local anomalous bodies, will be absorbed by the absorbing boundary condition, preventing the reflection phases of the scattered waves from returning to the interior of the simulation box, leading to a complete absence of second-order scattering energy for stations outside the box.

The formulation is extensively tested in computational experiments in 2 and 3D, both with the box containing the core-mantle boundary (CMB) and the ocean floor. In the 2D experiments, identical spatial meshes and time steps in the global and local numerical simulations are employed providing optimal accuracy. The "convergence" studies are focused on comparisons between hybrid and global solutions. I would like to suggest that the authors check the presence of Scholte waves, for example, in Figure 3.

- Thank You for this suggestion. For the Scholte waves in real data, we refer to Li et al., NC (2022) ("Kilometer-scale structure on the core-mantle boundary near Hawaii," <https://www.nature.com/articles/s41467-022-30502-5>). In Figure 4 of Li et al., 2022, the faded yellow shading represents the time delay ranges for the long- and short-period Sdiff postcursors, attributed to the presence of a kilometer-scale ULVZ as observed in the real data (Figures 3A and 3D). These seismic phases are analogous to surface waves. We also have added a few sentences to explain the new phases generated by the local structure above CMB (Page 6 and Line 147): "Note that the dispersed scattered surface waves are generated by the thin ULVZ anomaly above the CMB. In contrast, the Scholte wave, which has a lower velocity than the S wave, is produced due to the topography of the CMB. Additionally, an S-P scattered phase is generated when the S phase interacts with the topography."

METHODOLOGY:

- In general, the numerical approach the authors propose, in all its details, is correct. The acoustic-elastic system of equations is used to model seismic waves. I presume that the authors consider

relatively high-frequency waves and thus ignore self-gravitation. The self-gravitation affects the fluid-solid boundary in a significant way. I would like to suggest that the authors extend their discussion on related work.

- We have clarified this on Line 395. Indeed, due to the target of probing small-scale structures in the deep Earth, in this study, we only consider relatively low-period waves (8 to 20 s in 3D) and ignore self-gravitation. The self-gravitation is significant at longer periods (e.g. Chaljub and Valette, 2004). We mention this on lines 406-410.
- To my understanding, the earlier work by Johan Robertsson and Chris Chapman on "An efficient method for calculating finite-difference seismograms after model alterations" (2000; <https://library.seg.org/doi/epdf/10.1190/1.1444787>) is directly related to box tomography.
- This is a very important and relevant reference, thank you for your suggestion, we are now citing it, (Line 47).
- It has been recognized that fluid-solid boundaries cannot be dealt with within a purely spectral element method. In fact, such boundaries fit naturally a discontinuous Galerkin approach
 - Käser and Dumbser 2008, <https://library.seg.org/doi/10.1190/1.2870081>;
 - Ye et al. 2016, <https://academic.oup.com/gji/article/205/2/1267/691117>;
 - Guo et al. 2020, <https://www.sciencedirect.com/science/article/pii/S002199912030406X>.

Nonetheless, there is a "fix" for spectral elements in the neighborhood of a fluid-solid boundary.

- Line 363. Both the Discontinuous Galerkin Method (DGM) and the recently proposed Distributional Finite Difference Method (DFDM) naturally handle fluid-solid boundaries by directly setting the shear modulus to zero. This makes the implementation of hybrid solid-fluid coupling in both DGM and DFDM promising.
- An early reference to the use of velocity potential and the Finite Element Method is Olson and Bathe, "Analysis of fluid-structure interactions. A direct symmetric coupled formulation based on the fluid velocity potential" (1985, <https://www.sciencedirect.com/science/article/pii/0045794985902263>). By changing the definition of displacement potential, one might think that the authors might lose the symmetry of the discrete formulation. It would be helpful to comment on this and explain which inner product / Hilbert space is used in the fluid-solid configuration (equations (12)-(13)).
- As you noted, losing symmetry in the formulation could result in an unstable scheme when applied in the time domain.

The numerical simulation of acoustic and elastic equations, along with the continuous interface conditions of normal displacement and traction at the solid-fluid coupling interface, has been carefully selected and performed to maintain stability. In the discrete system (equation (16)) we constructed for equations (12) and (13), the mass matrices for both the elastic wave equation and the acoustic wave equation are diagonal and inherently symmetric. A second-order central difference method is used in the time direction. The process involves:

- (i) first updating the displacement and displacement potential at time step $n+1$ using the known displacement and displacement potential from time steps $n-1$ and n , as well as the solid and fluid accelerations at time step n ;
- (ii) calculating the fluid acceleration at time step $n+1$ using the displacement at the same time step;
- (iii) calculating the solid acceleration at time step $n+1$ using the fluid acceleration at that step.

All operations related to the hybrid simulation involve only adding precomputed mirror forces, so they do not impact the stability of the solid-fluid coupling. To further analyze the stability of the solid-fluid coupling, first, let's define the global unknown vector, global mass matrix, and global stiffness matrix as follows:

$$\mathbf{U} = \begin{bmatrix} \ddot{\Phi}_f \\ \ddot{\Phi}_b \\ \ddot{\mathbf{U}}_b \\ \ddot{\mathbf{U}}_s \end{bmatrix}, \quad (1)$$

$$\mathbf{M}^g = \begin{bmatrix} \mathbf{M}_{ff} & \mathbf{M}_{fb} & 0 & 0 \\ \mathbf{M}_{bf} & \mathbf{M}_{bb}^f & 0 & 0 \\ 0 & \mathbf{A}^T & \mathbf{M}_{bb}^s & \mathbf{M}_{bs} \\ 0 & 0 & \mathbf{M}_{sb} & \mathbf{M}_{ss} \end{bmatrix}, \quad (2)$$

and

$$\mathbf{K}^g = \begin{bmatrix} \mathbf{K}_{ff} & \mathbf{K}_{fb} & 0 & 0 \\ \mathbf{K}_{bf} & \mathbf{K}_{bb}^f & -\mathbf{A} & 0 \\ 0 & 0 & \mathbf{K}_{bb}^s & \mathbf{K}_{bs} \\ 0 & 0 & \mathbf{K}_{sb} & \mathbf{K}_{ss} \end{bmatrix} \quad (3)$$

which is an extended version of Equation (16) in the main text. Here Φ , \mathbf{U} , \mathbf{M} , and \mathbf{K} are displacement potential, displacement, mass matrix, and stiffness matrix. The subscripts f , b , and s represent the fluid domain, solid-fluid coupling boundary, and the solid domain, respectively.

Next, we can construct the global system

$$-\mathbf{M}^g \ddot{\mathbf{U}} = \mathbf{K}^g \mathbf{U}, \quad (4)$$

We observe that the symmetric positive global mass matrix has been altered by the solid-fluid coupling boundary, with the vector \mathbf{A}^T , and the symmetric global stiffness matrix is also modified by the vector \mathbf{A} .

Then, we can rewrite the solved system as below:

$$-(\mathbf{U}^{n+1} - 2\mathbf{U}^n + \mathbf{U}^n) = \underbrace{\Delta t^2 (\mathbf{M}^g)^{-1} \mathbf{K}^g}_{\mathbf{B}} \mathbf{U}^n. \quad (5)$$

The left side of equation ?? in the frequency domain

$$-(e^{iw\Delta t} - 2 + e^{-iw\Delta t})\tilde{\mathbf{U}} = 4\sin^2(w\Delta t/2)\tilde{\mathbf{U}}. \quad (6)$$

Here, $\tilde{\mathbf{U}} = F^+(\mathbf{U}(t))$, where F^+ denotes the forward Fourier transform. The coefficient range of $\tilde{\mathbf{U}}$ should lie within $[0, 4]$. The matrix \mathbf{B} must be a semi-positive definite matrix, like the updated matrix in FD and SEM, and should only have non-negative real eigenvalues, not greater than 4, to ensure the system's stability (Gaffar and Jiao, 2014; Gao et al., 2018; Lyu et al., 2021).

- Gaffar and Jiao, 2014: <https://ieeexplore.ieee.org/document/6910330>;
- Gao et al., 2018: <https://doi.org/10.1190/geo2018-0447.1>;
- Lyu et al., 2021: <https://doi.org/10.1190/geo2020-0623.1>.

Table 1: Eigenvalues values of solid-fluid coupling case

1.805601549	1.795402893	1.477626273	1.387959241	1.314306687	1.251040204	1.182982015
1.091228613	1.035914897	1.034391668	0.983512580	0.969654258	0.890598433	0.903861083
0.822847427	0.744627750	0.713075551	0.558033077	0.538308501	0.517231269	0.493150966
0.474431514	0.481203634	0.432093676	0.416035118	0.395375506	0.378253914	0.363742141
0.352863924	0.336480619	0.334946937	0.316474198	0.309621777	0.292824034	0.280831653
0.255638246	0.250707013	0.242959705	0.202654955	0.196078775	0.193259849	0.184888254
0.190644824	0.171914795	0.175258082	0.158437284	0.154059733	0.132177684	0.120554600
0.110618259	0.001025405	0.008132930	0.007506173	0.015601147	0.018847902	0.020103407
0.022569021	0.027812798	0.031266074	0.035647283	0.041112399	0.046083761	0.054126038
0.057310077	0.096059324	0.093276614	0.088698586	0.065774178	0.083391723	0.070632678
0.072046510	0.078091718	0.000000000	0.000000000	0.000000000		

Since the inverse of the mass matrix exists in this solution system, it is challenging to directly prove theoretically that matrix B is a semi-positive definite matrix. However, we can obtain matrix B directly through an implicit method (by assigning a value of 1 to each degree of freedom one by one, while assigning 0 to the other degrees of freedom). We tested many different solid-fluid coupling models, and all the eigenvalues of the matrix B are non-negative real eigenvalues. These numerical tests demonstrate that the solid-fluid coupling system is stable.

Here, we provide a 2D solid-fluid coupling numerical simulation and list the corresponding eigenvalues. This 2D model is configured as follows: two elements are positioned above and below the CMB, which serves as the common boundary between them. Each element contains 25 GLL points, resulting in a 75 by 75 matrix B . The parameters for the lower mantle are $V_p = 13,716.62$ m/s; $V_s = 7,264.65$ m/s; and the density = $5,566.46$ kg/m³. For the fluid outer core, $V_p = 8,064.79$ m/s; and the density = $9,903.44$ kg/m³. The horizontal length is set to 0.125 degrees, and the vertical thickness is 10 km for both the mantle and the outer core. A time step $\Delta t = 0.04$ s is used. The final 75 eigenvalues of matrix B are listed in Table 1.

We are including this point in Section 1 of the Supplementary Information, and adding "For the verification of the stability of the solid-fluid coupling, refer to Section 1 of SI." in the main text on Line 433.

- In equation (13), in the spectral element formulation of Chaljub and Valette (2004), the average of the fluxes on the fluid-solid boundaries is taken; why did the authors choose not to do that?
- To focus on physical effects that are critical for the lower frequency band (≥ 100 s), Chaljub and Valette (2004) decomposed the displacement in the fluid domain into two scalar displacement potentials (χ and ξ). In this study, we focus on relatively lower periods (≤ 20 s) and have neglected self-gravitation, resulting in a simpler wave equation. Both approaches result in a fully explicit fluid-solid coupling strategy. The key differences in implementations are as follows.

(1). In Chaljub and Valette (2004), displacement in the fluid domain is defined by their Eq. (8): $\mathbf{u} = \nabla\chi + \xi\mathbf{s}$ with the definition Eq. (9) $\xi = c^2\nabla \cdot \mathbf{u}$. Based on Equations (3, 8, 9), the continuity of traction across the solid-fluid boundaries is implemented in their Eq. (18) by replacing the traction on the solid-fluid boundaries from the solid side with $\check{\xi}$ in the fluid domain ($\mathbf{T}(\mathbf{u}) = \rho c^2\nabla \cdot \mathbf{u}\mathbf{I} = \rho\check{\xi}\mathbf{I}$). The continuity of displacement across the solid-fluid boundaries is implemented as shown in equation (19) by replacing the displacement on the solid-fluid boundaries from the fluid side with displacement from the solid domain ($\nabla\chi + \xi\mathbf{s} = \mathbf{u}$).

(2). In our study, we use a single displacement potential (Eqs. (4) and (7)) in our manuscript: $\mathbf{u}_f = \frac{1}{\rho_f}\nabla\varphi$ (2D) or $\mathbf{u}_f = \frac{1}{\rho_f}\nabla(\rho_f\phi)$ (3D)), and the corresponding solid-fluid coupling is implemented in equations (13) and (15). To ensure continuity of traction and displacement across the

solid-fluid coupling, in 2D, equation (13) replaces the traction $\mathbf{T} = \boldsymbol{\sigma} \cdot \mathbf{n}$ by $\ddot{\varphi}$ and the displacement \mathbf{u} by $\frac{1}{\rho_f} \nabla \varphi$. In 3D, equation (15) replaces the traction $\mathbf{T} = \boldsymbol{\sigma} \cdot \mathbf{n}$ by $\rho_f \ddot{\phi}$, and replaces the displacement \mathbf{u} by $\frac{1}{\rho_f} \nabla(\rho_f \phi)$

From the comparison above, we can see that our method, especially in 3D, is quite similar to that of Chaljub and Valette (2004). However, because of no gravity effects, our equations are simpler, only one displacement potential is needed. See added text: Page 16 and Line 406.

- As the details of the spectral elements are omitted (which is very reasonable), it seems that the entire subsection entitled "Mathematical expression of hybrid input and output mirror forces" can be omitted.
- In addressing the challenge of hybrid solid-fluid coupling, we believe that the precise expression of the hybrid input and output mirror force equations is essential. While our formulation is based on the spectral element method (SEM), the same format is applicable to other numerical methods, such as finite differences, which also incorporate hybrid numerical techniques (see, for example, section 3.1.2 in Masson et al., 2014 GJI, <https://doi.org/10.1093/gji/ggt459>), albeit with a simpler mass matrix. Therefore, we have decided to retain these equations and explicitly mention in the text that they are derived using SEM. Nonetheless, the underlying mathematical and physical principles remain applicable to other numerical methods, including finite difference methods (FDM).

SIGNIFICANCE:

- The significance of the approach presented in the paper is in making the high-resolution study of the CMB region computationally feasible. Rather than resorting to traditional gradient methods, I imagine that plug and play (PnP, see, for example, the work of Ulugbek Kamilov) and the use of image denoisers will find application in deep interior imaging with seismic waves.
- Thank you for your suggestion. We have added it on Line 347. Indeed, our ultimate goal is to significantly reduce the time required for forward modeling, thereby bypassing traditional kernel methods and employing more advanced techniques for high-resolution imaging of the Earth's deep interior. These techniques include machine learning, as well as the plug and play (PnP) and image denoisers mentioned by you (e.g., Ulugbek Kamilov, 2021).

Efficiently simulating seismic wave propagation in full 3-D structures is crucial for accurate seismic imaging of Earth's interior, especially for applications like full waveform inversion. Despite advancements such as AxiSEM, which can model smooth 3-D structures, simulating high-frequency seismic waves globally remains challenging. To address this, the authors in this study introduce a hybrid method for box tomography, significantly reducing computational costs while maintaining accuracy. By conducting global simulations once and limiting multiple expensive simulations to smaller local domains, the authors demonstrate that the computational cost could be reduced by three orders of magnitude. Specifically, the authors focus on the examples of solid-fluid coupling simulations in this study and have conducted well-designed benchmarking against full 3-D simulations. This novel method has tremendous potential of widespread applications, including detailed imaging of Earth's interior features like the oceanic lithosphere, mantle transition zone, lower mantle, and inner core. Any progress on these topics will produce impact beyond seismology. Thus, I highly recommend publishing this study on Nature Communications with some minor revisions as described below.

Contributions from different error sources. The authors discussed errors from spatial and temporal interpolations and Absorbing Boundary Conditions (ABC). However, a deeper analysis of the contributions from each error source would provide valuable insights. Analyzing and comparing the errors due to different model setups can tell us the contribution of each error source. For example, the results in Fig. 4 uses the same mesh, so the errors must come from imperfect ABC; In Fig. 8c,d of SORO case, which error between these sources is larger? If the primary error is from imperfect ABC, better ABCs would help reduce the error. Conversely, if interpolation errors dominate, is there a solution to reduce the error? Further discussions would enhance the manuscript's comprehensiveness and guide future research endeavors.

Meshing ULVZ. In the 3-D ULVZ simulations, does the ULVZ in Fig. 7 have smoothed structure anomalies? The authors mentioned a smooth ULVZ in Fig. S9 and S10, and I suppose it is the same in Fig. 7. It should be clarified. In addition, it is worth of noting that one advantage of hybrid method is that the flexible meshing of local domain allows us to better honor the geometry of ULVZ with sharp boundaries, which would be difficult for global meshing.

The waveform details in Fig. 8 suffer from visibility issues due to the plotting scale, hindering the comprehensive understanding of the results. To address this limitation, I recommend replotting Fig. 8 to better visualize the detailed waveforms.

$\int_{\Omega_s} w \rho_s \ddot{u}_s d\Omega_s$ in eq. 12 should be $\int_{\Omega_s} \rho_s w \cdot \ddot{u}_s d\Omega_s$. The product symbol “.” is also missing in the last term of eq. 12 and many other equations (eqs. 13, 14, 15 and 19).

For eq. 16, add clarification that the corresponding absorbing boundary conditions are Stacey ABC.

Clarify what types of sources (explosion, double-couple or what else?) are used in the 2-D (i.e., Fig. 3) and 3-D simulations (i.e., Fig. 7).

The colors and plotting in Fig. 4 warrant clarification to improve understanding. I think Fig. 4a,b plots scattering waves, which should be theoretically zeros, as the authors described. But how about Figs. 4c-f? Are they solely scattering waves, or do they combine scatter waves with background 1-D waveforms? If they are scattering waves, it is better to use consistent color in (a)-(f) for hybrid seismograms. Currently, hybrid seismogram is black in a,b while red in c-f. Furthermore, plotting both background 1-D seismograms and scattering waves for comparison would enhance understanding of ULVZ and CMB topography effects. This comparative visualization would elucidate how these heterogeneities influence wave propagation, providing deeper insights into the study's findings.

Describe the correspond time of the snapshots in Fig. 3b,d,f and Fig. 5b,d. That would help us better link them with the corresponding seismograms.

Line 196: replace “minimum period” with “minimum resolved period”. And revise it at other relevant places as well.

There are many interesting waveform propagation phenomena in Fig. 3 deserving further analysis. For example, it would be helpful to label out the arrivals S and ScS in Fig. 3b; Explaining the scattering waves, as marked in the below figure, would offer valuable insights into how these waves are generated by the structural heterogeneities.

In Section 1 of the support material, the authors justify the use of acceleration potential rather than potential in equation S8 through dimension analysis. This justification is fine, but this can be better explained by revisiting the governing equations. Given the ‘force’ formulas of S5 for the scenario of source to SEM box coupling, one would write an analogous formula for the coupling from the SEM box to a remote receiver. Indeed, If SEM is utilized to back propagate the imposed wavefield on the A_2 boundaries to a

remote receiver, employing the potential is appropriate. However, if Green's functions are chosen for convolution, we should use a real 'force' term, whereas F_f^{hi} in equation S5 is actually not a 'force'. Although F_f^{hi} is termed a 'force' due to its analogy to its solid media counterpart F_s^{hi} , this analogy exists in their mathematical representations. Physically, they are different (they have different units). Similarly, the two equations of S3 have analogous mathematical representation, but not in physics. Equations 1 and 2, along with their counterparts f_s and f_f , exhibit a more direct analogy in both mathematics and physics. Hence, if convolution is employed to derive the synthetic seismogram at a remote station, using equivalent body 'forces' akin to f_f would be natural, which differs from F_f^{hi} . Consequently, F_f^{hi} does not have a unit of 'force' and referring to F_f^{hi} as a 'force' is somewhat inaccurate in this context. By providing this clarification, the physical and mathematical nuances between F_f^{hi} and equivalent body forces become apparent.

In Section 4 of the support material, it's mentioned that the density of the Ultra-Low Velocity Zone (ULVZ) is set as increasing by 14.3%. However, it's essential to clarify that this value is utilized solely for synthetic testing purposes, and we know that observational data supports a higher density for ULVZ.